# DILATED CONVOLUTION WITH LEARNABLE SPACINGS

**Ismail Khalfaoui-Hassani**
Artificial and Natural Intelligence Toulouse Institute (ANITI)
Université de Toulouse, France
`ismail.khalfaoui-hassani@univ-tlse3.fr`

**Thomas Pellegrini**
IRIT, ANITI, Université de Toulouse,
CNRS, Toulouse INP, UT3, France
`thomas.pellegrini@irit.fr`

**Timothée Masquelier**
CerCo UMR 5549
CNRS & Université de Toulouse, France
`timothee.masquelier@cnrs.fr`

## ABSTRACT

Recent works indicate that convolutional neural networks (CNN) need large receptive fields (RF) to compete with visual transformers and their attention mechanism. In CNNs, RFs can simply be enlarged by increasing the convolution kernel sizes. Yet the number of trainable parameters, which scales quadratically with the kernel's size in the 2D case, rapidly becomes prohibitive, and the training is notoriously difficult. This paper presents a new method to increase the RF size without increasing the number of parameters. The dilated convolution (DC) has already been proposed for the same purpose. DC can be seen as a convolution with a kernel that contains only a few non-zero elements placed on a regular grid. Here we present a new version of the DC in which the spacings between the non-zero elements, or equivalently their positions, are no longer fixed but learnable via backpropagation thanks to an interpolation technique. We call this method "Dilated Convolution with Learnable Spacings" (DCLS) and generalize it to the n-dimensional convolution case. However, our main focus here will be on the 2D case for computer vision only. We first tried our approach on ResNet50: we drop-in replaced the standard convolutions with DCLS ones, which increased the accuracy of ImageNet1k classification at iso-parameters, but at the expense of the throughput. Next, we used the recent ConvNeXt state-of-the-art convolutional architecture and drop-in replaced the depthwise convolutions with DCLS ones. This not only increased the accuracy of ImageNet1k classification but also of typical downstream and robustness tasks, again at iso-parameters but this time with negligible cost on throughput, as ConvNeXt uses separable convolutions. Conversely, classic DC led to poor performance with both ResNet50 and ConvNeXt. The code of the method is based on PyTorch and available.

## 1 INTRODUCTION

The receptive field of a deep convolutional network is a crucial element to consider when dealing with recognition and downstream tasks in computer vision. For instance, a logarithmic relationship between classification accuracy and receptive field size was observed in Araujo et al. (2019). This tells us that large receptive fields are necessary for high-level vision tasks, but with logarithmically decreasing rewards and thus a higher computational cost to reach them.

Recent advances in vision transformers (Dosovitskiy et al., 2020) and in CNNs (Liu et al., 2022b; Ding et al., 2022; Trockman & Kolter, 2022; Liu et al., 2022a) highlight the beneficial effect that a large convolution kernel can have, compared to the $3 \times 3$ kernels traditionally used in previous state-of-the-art CNN models (He et al., 2016). However, when naively increasing the kernel size, the accuracy rapidly plateaus or even decreases. For example, in ConvNeXt, the best accuracy was achieved by a $7 \times 7$ kernel (Liu et al., 2022b;a). Using a structural re-parameterization trick, Ding et al. (2022) demonstrated the benefit of increasing the kernel size up to 31 by 31. Thereafter, Liu et al. (2022a) showed that there was still room for improvement by moving to 51 by 51, us-

ing the *depthwise implicit matrix multiplication (gemm)* method developed by Ding et al. (2022) and for which the implementation has been integrated into the open-sourced framework MegEngine (Megvii, 2020), in addition to a spatial separation of the depthwise kernel followed by an accumulation of the resulting activations. Yet, all these improvements have a cost in terms of memory and computation, and it does not seem possible to increase the size of the kernels indefinitely.

One of the first approaches that allow inflating the receptive field of a convolutional layer without increasing the number of learnable parameters nor the computational cost is called dilated convolution (DC). DC or "atrous convolution" was first described in Holschneider et al. (1990) and Shensa (1992), under the name "convolution with a dilated filter" before being referred to as "dilated convolution" in Yu & Koltun (2015). The purpose of this approach is to inflate the convolutional kernel by regularly inserting spaces (*i.e.* zeros) between the kernel elements, as depicted in Figure 2b. The spacing between elements is thus constant, it is a hyper-parameter usually referred to as "dilation" or "dilation rate". Despite its early successes in classification since Yu et al. (2017), and its even

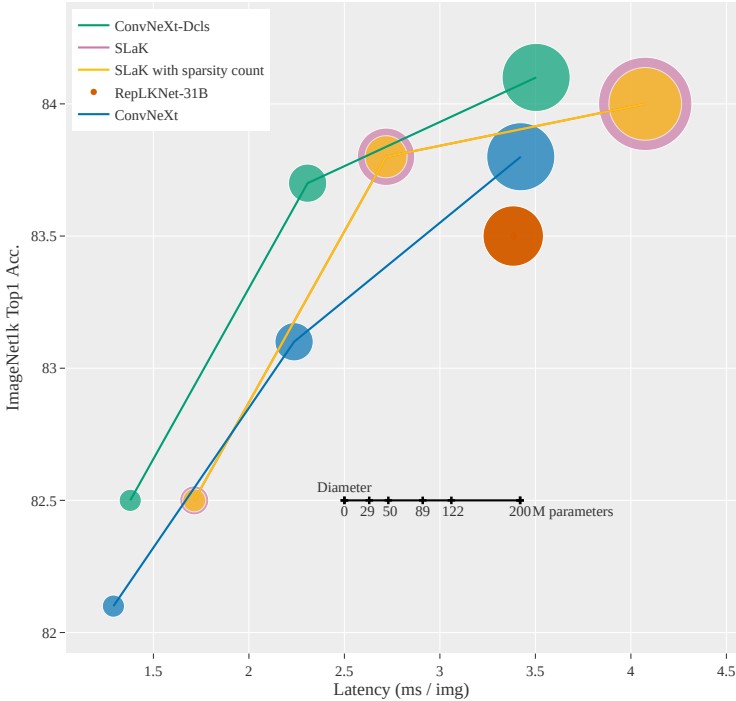

Figure 1: Classification accuracy on ImageNet-1K as a function of latency (i.e. inverse of the throughput). Dot diameter corresponds to the number of parameters.

more convincing results in semantic segmentation Sandler et al. (2018); Chen et al. (2017; 2018) and object detection Lu et al. (2019), DC has gradually fallen out of favor and has been confined to downstream tasks such as those described above. Without much success, Ding et al. (2022) tried to implement DC in their ReplKNet architecture. Our own investigation on ResNet and ConvNeXt with standard dilated convolution (Section 4.2) will lead to a similar conclusion. The failure of this method for classification tasks could be attributed to the great rigidity imposed by its regular grid as discussed in Wang & Ji (2018).

In this context, we propose DCLS (Dilated Convolution with Learnable Spacings), a new convolution method. In DCLS, the positions of the non-zero elements within the convolutional kernels are learned in a gradient-based manner. The inherent problem of non-differentiability due to the integer nature of the positions in the kernel is circumvented by interpolation (Fig. 2c). DCLS is a differentiable method that only constructs the convolutional kernel. To actually apply the method, we could either use the native convolution provided by PyTorch or a more advanced one such as the *depthwise implicit gemm* convolution method (Ding et al., 2022), using the constructed kernel. DCLS comes in six sub-versions: 1D, 2D, 3D and what we call N-MD methods, namely: "2-1D, 3-1D and 3-2D" where a N-dimension kernel is used but positions are learned only along M dimension(s). The main

focus of this paper will be the 2D version for which we detail mathematical proofs, implementation specificities and results on image classification, downstream and robustness tasks.

The principal motivation of DCLS is to explore the possibility of improving the fixed grid imposed by the standard DC via learning the spacings in an input-independent way. Instead of having a grid of kernel elements like in standard and dilated convolutions, DCLS allows an arbitrary number of kernel elements (Fig. 2d). We refer to this free tunable hyper-parameter as "kernel count". In this paper, we set it in order to be at iso or fewer parameters than the baselines we will compare ourselves to. Conversely, we refer to the size of the kernel, or rather the maximum size in which the kernel elements are allowed to move inside the dilated kernel, as the "dilated kernel size". It is also a tunable hyper-parameter.

The positions of kernel elements in DCLS are randomly initialized, and are allowed to move throughout the learning process within the dilated kernel size limit. We will then show how sharing positions across multiple blocks of a same convolution stage could further increase accuracy while reducing the number of learnable parameters. This, together with other learning techniques, empirically and consistently improve the overall performance of the method. They are summarized in Section 3.

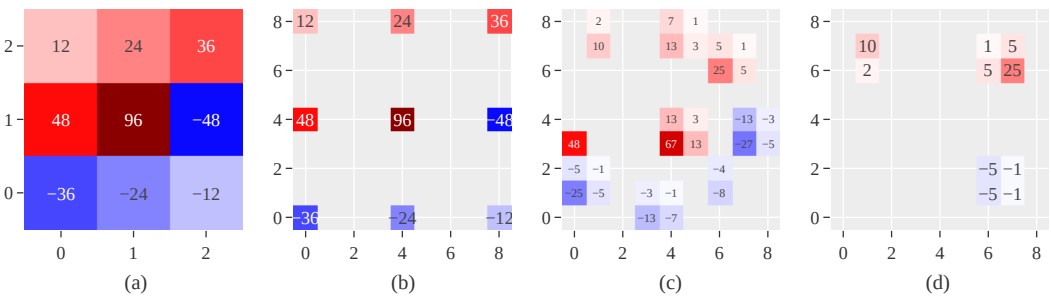

Figure 2: (a): a standard $3 \times 3$ kernel. (b): a dilated $3 \times 3$ kernel with dilation rate 4. (c): a 2D-DCLS kernel with 9 kernel elements and a dilated kernel size of 9. Each weight is spread over up to four adjacent pixels. (d): a 2D-DCLS kernel with 3 kernel elements and still a dilated kernel size of 9.

## 2 KERNEL CONSTRUCTION IN DCLS

Our method entails that we learn the float coordinates ($p^1$ and $p^2$ in the 2D case for example) for every weight $w$ of the dilated kernel (in addition to the actual weights themselves). The positions or coordinates of the weights within the convolution kernel are conceptually integer parameters, yet in order to compute their derivatives, we should consider them as float parameters. This problem of integer positions learning is smoothed by making use of a bilinear interpolation that will be described in equations 2, 3 and 4.

### 2.1 NOTATION AND PRELIMINARIES

We denote by $\lfloor\ \rfloor$ the floor function and we define its derivative by the zero function.

$$\forall x \in \mathbb{R}, \lfloor x \rfloor' \overset{\text{def}}{=} 0 \tag{1}$$

We denote by $m \in \mathbb{N}^*$ the number of kernel elements inside the constructed kernel and we refer to it as the "kernel count". Moreover, we denote respectively by $s_1, s_2 \in \mathbb{N}^* \times \mathbb{N}^*$, the sizes of the constructed kernel along the x-axis and the y-axis. The latter could be seen as the limits of the dilated kernel, and we refer to them as the "dilated kernel size".

The $s_1 \times s_2$ matrix space over $\mathbb{R}$ is defined as the set of all $s_1 \times s_2$ matrices over $\mathbb{R}$, and is denoted $\mathcal{M}_{s_1,s_2}(\mathbb{R})$.

The characters $w$, $p^1$ and $p^2$ respectively stand for the weight, the position of that weight along the x-axis (width) and its position along the y-axis (height) in the scalar case while the bold $\boldsymbol{w} =$

$(w_i)_{1 \leq i \leq m}$, $\boldsymbol{p}^1 = (p_i^1)_{1 \leq i \leq m}$ and $\boldsymbol{p}^2 = (p_i^2)_{1 \leq i \leq m}$ respectively stand for the weight, the width-position of that weight and its height-position in the vector case.

## 2.2 MATHEMATICAL FORMULATION

The mathematical construction of the forward pass as well as the proofs for the derivations used in the backward pass are in Appendix 7. This construction relies on bilinear interpolation and could be described by the following function:

$$
\begin{aligned}
F: \quad & \mathbb{R}^m \times \mathbb{R}^m \times \mathbb{R}^m \to \mathcal{M}_{s_1, s_2}(\mathbb{R}) \\
& \boldsymbol{w}, \boldsymbol{p}^1, \boldsymbol{p}^2 \mapsto \boldsymbol{K} = \sum_{k=1}^m f(w_k, p_k^1, p_k^2)
\end{aligned}
\tag{2}
$$

With $f$ defined as follows:

$$
\begin{aligned}
f: \mathbb{R} \times \mathbb{R} \times \mathbb{R} &\to \mathcal{M}_{s_1, s_2}(\mathbb{R}) \\
w, p^1, p^2 &\mapsto \quad \boldsymbol{K}
\end{aligned}
\tag{3}
$$

where $\forall i \in [\![ 1 .. s_1 ]\!], \forall j \in [\![ 1 .. s_2 ]\!]$:

$$
\boldsymbol{K}_{ij} = \begin{cases}
w \left(1 - r^1\right) \left(1 - r^2\right) & \text{if } i = \lfloor p^1 \rfloor, \ j = \lfloor p^2 \rfloor \\
w \, r^1 \left(1 - r^2\right) & \text{if } i = \lfloor p^1 \rfloor + 1, \ j = \lfloor p^2 \rfloor \\
w \left(1 - r^1\right) r^2 & \text{if } i = \lfloor p^1 \rfloor, \ j = \lfloor p^2 \rfloor + 1 \\
w \, r^1 \, r^2 & \text{if } i = \lfloor p^1 \rfloor + 1, \ j = \lfloor p^2 \rfloor + 1 \\
0 & \text{otherwise}
\end{cases}
\tag{4}
$$

and where the fractional parts are:

$$
r^1 = \{p^1\} = p^1 - \lfloor p^1 \rfloor \quad \text{and} \quad r^2 = \{p^2\} = p^2 - \lfloor p^2 \rfloor
\tag{5}
$$

We invite the reader to look at Appendix 7 for more details on the mathematical proofs and derivatives that permits to calculate the gradients of a differentiable loss function with respect to the weights and their positions. Those derivations lead to the DCLS kernel construction algorithm described for the 2D case in pseudo-code in Appendix 8. Furthermore, the real code for the kernel construction in 1D, 2D, and 3D cases is included in Appendix 9. This code is written in native PyTorch language, with classical modules, and does not require any compilation or adaptation.

## 3 LEARNING TECHNIQUES

So far, we have seen how to implement the DCLS method. We now turn to the techniques that allow us to get the most out of the method. In what follows, we list the training techniques that we have retained and for which we have noticed a consistent and empirical advantage on validation accuracy.

- **Weight decay:** weight decay is a regularization method widely used in a variety of deep learning models. Though its beneficial effect on generalization, we noticed that when applied to the kernel positions in DCLS method, weight decay tends to "artificially" over-concentrate the positions around the center of the kernel, resulting in poorer accuracy. Therefore, we set this hyperparameter to 0 for the kernel positions and kept it unchanged for all the other parameters.

- **Positions initialization:** the DCLS positions tend to cluster around the center of the RF throughout the learning process (see Appendix 10). In an attempt to facilitate learning, we chose an initial distribution close to the one obtained at the end of training, that is a centered normal law of standard deviation 0.5. Yet in practice, the uniform law gives a similar performance.

- **Positions clamping / overlapping:** kernel elements that reach the dilated kernel size limit are clamped. This is done at the end of every batch step to force kernel positions to stay within limits. Agglutination around those limits can sometimes be observed and this indicates that the dilated kernel size is too small and should be enlarged. Positions of different kernel weights (or their interpolations) could also overlap. They are added together in such a case.

- **Dilated kernel size tuning:** we empirically tuned it using the remark above. For simplicity, we used the same dilated kernel size in all the model layers (7 for ResNet-50-dcls and 17 for ConvNeXt-dcls; larger values did not bring any gain in accuracy). Note that increasing the dilated kernel size has no cost on the number of trainable parameters, but has a cost on throughput, especially when using non-separable convolutions. Besides, the convolution algorithm (which is the most time-consuming part of DCLS) that is used after the kernel construction (whether it is the native one or the *depthwise implicit gemm* one) does not leverage the kernel sparsity. Thus, the kernel count does not impact the throughput; only the dilated kernel size does. The development of a convolution method that takes into account sparse kernels, such as Kundu et al. (2019), could further help to reduce the throughput gap between DCLS convolution and standard dilated convolution.

- **Kernel count tuning:** as said before, we have set this hyper-parameter to the maximal integer value that allows us to be below the baselines to which we compare ourselves in terms of the number of trainable parameters. Note that adding one element to the 2D-DCLS kernel leads to having three additional learnable parameters: the weight, its vertical and its horizontal position. For simplicity, we used the same kernel count in all the model layers.

- **Positions learning rate scaling:** we found that kernel positions could benefit from a special scaling when it comes to the learning rate. As their magnitude is different from regular weights, we scaled the learning rate of all the kernel positions by a factor of 5. This is the best factor we found empirically. Custom-made schedulers for positions have been tested, but scaling the learning rate while using the same scheduler as for the kernel weights remained the best choice. Interestingly, we found that throughout learning, the average speed of the positions follows precisely the shape of the learning rate scheduler curve (see Appendix 11).

- **Synchronizing positions:** we shared the kernel positions across convolution layers with the same number of parameters (typically, those belonging to a same stage of the ConvNeXt/ResNet model), without sharing the weights. Positions in this kind of stages were centralized in common parameters that accumulate the gradients. This constraint has surprisingly enhanced the accuracy while reducing the number of extra parameters dedicated to positions (an ablation of this technique led to a 0.13% accuracy drop on ImageNet1k with ConvNeXt-T-dcls).

- **Repulsive loss**: following the work of Thomas et al. (2019) on 3D cloud points, we implemented the repulsive loss for the DCLS kernel positions to discourage multiple elements from overlapping. Despite a slight advantage with the ResNet-50-dcls model, this technique did not significantly improve the results with ConvNeXt-dcls.

- ***Depthwise implicit gemm***: This method has been developed by Ding et al. (2022) and integrated in Megvii (2020), it has for goal to modify the *im2col* algorithm as well as the matrix multiplication that follows, both used in the native 2D convolution method of PyTorch, by a very efficient algorithm which does not build explicitly the *im2col* tensor. The advantage of this method is that it is much faster than the native one for large kernel sizes, without affecting the convolution speed for small kernel ones. In our experiments, the largest dilated kernel size is 17, therefore this method is not absolutely necessary. However, the DCLS user can choose to use it instead of the native method, which will improve the throughput.

## 4 RESULTS AND DISCUSSION

### 4.1 SETTINGS

We started with an exploratory study on ResNet-50, where we drop-in replaced all the $3 \times 3$ convolutions of the model by 2D-DCLS ones. For that, we used a lightweight procedure named the "A3 configuration", described in Wightman et al. (2021). We then moved to the ConvNeXt models, where we limited our studies to its three first variants namely: the tiny, the small and the base models with input crops of size $224 \times 224$ (Liu et al., 2022b). Here, we drop-in replaced all the depthwise convolutions of the model by DCLS ones. We reconducted all the experiments, and evaluated the seed sensitivity for the ConvNeXt-dcls model by calculating the standard deviation of the top-1 accuracy on three different seeds, for the tiny variant. We found the standard deviation to be $\pm 0.04$, which is compatible with what was found in Liu et al. (2022b). Given this reasonably low variability, the remaining experiments were done on one seed only. Code and scripts to reproduce the training are available (see reproducibility statement 6).

## 4.2 EMPIRICAL EVALUATIONS ON IMAGENET1K

In the following, we report the top-1 accuracies found on the ImageNet1k validation dataset (Deng et al., 2009), using ImageNet1k training dataset only.

**Using ResNet-50.** Table 1 presents the results obtained for the ResNet-50 experiment using the A3 configuration. The aim of this study is not to exceed the state-of-the-art for this particular configuration, but rather to give a first experimental evidence of the relevance of the DCLS method with non-separable convolutions and with one of the most popular CNN architectures in computer vision. We can observe that when using the standard dilated convolution, the results only get worse as we increase the dilation rate. Moreover, increasing the kernel size ($3 \rightarrow 7$) in the case of standard convolution increases the accuracy but at the expense of not only the throughput, which decreases, but also of the number of parameters, which triples.

With fewer parameters, the ResNet-50-dcls model surpasses the baseline but at the expense of the throughput. This last disadvantage is due to the fact that ResNet-50 uses non-separable convolutions. We will see in Table 2 that for the ConvNeXt model, this extra cost in throughput is minimized thanks to the depthwise separable convolution.

| model | kernel size / count | dil | # param. | FLOPs | throughput (image / s) | Top-1 acc. (crop 160) |
|---|---|---|---|---|---|---|
| ResNet-50 | 3/9 | 1 | 25.6M | 4.1G | **1021.9** | 75.8 |
| ResNet-50 | 7/49 | 1 | 75.9M | 12.3G | 642.6 | **77.0** |
| ResNet-50 | 3/9 | 2 | 25.6M | 4.1G | 931.8 | 71.7 |
| ResNet-50 | 3/9 | 3 | 25.6M | 4.1G | 943.4 | 70.1 |
| ResNet50-dcls ● | 7/5 | — | 24.0M | 12.3G | 627.2 | 76.5 |
| ResNet50-dcls ● | 7/6 | — | 26.0M | 12.3G | 627.1 | 76.5 |

Table 1: **Classification accuracy on ImageNet-1K using ResNet-50.** The throughput was calculated at inference time, on image crops of size $224 \times 224$ using a single V100-32gb gpu. When the model contains DCLS convolutions, we reported the kernel count and dilated kernel size. Otherwise, the kernel size is reported and thus the kernel count is in fact the square of that parameter.

| model | img size | # param. | FLOPs | throughput (image / s) | Top-1 acc. |
|---|---|---|---|---|---|
| Swin-T | $224^2$ | 28M | 4.5G | 757.9 | 81.3 |
| ConvNeXt-T ● | $224^2$ | 29M | 4.5G | **774.7** | 82.1 |
| ConvNeXt-T-dil2 | $224^2$ | 29M | 4.5G | **773.6** | 80.8 |
| ConvNeXt-T-ker17 | $224^2$ | 30M | 5G | 560.0 | 82.0 |
| SLaK-T ● ● | $224^2$ | 30● / 38●M | 5.0● / 9.4●G | 583.5 | **82.5** |
| ConvNeXt-T-dcls ● | $224^2$ | 29M | 5.0G | 725.3 | **82.5** |
| Swin-S | $224^2$ | 50M | 8.7G | 436.7 | 83.0 |
| ConvNeXt-S ● | $224^2$ | 50M | 8.7G | **447.1** | 83.1 |
| SLaK-S ● ● | $224^2$ | 55● / 75●M | 9.8● / 16.6●G | 367.9 | **83.8** |
| ConvNeXt-S-dcls ● | $224^2$ | 50M | 9.5G | 433.4 | **83.7** |
| Swin-B | $224^2$ | 88M | 15.4G | 286.6 | 83.5 |
| ConvNeXt-B ● | $224^2$ | 89M | 15.4G | **292.1** | 83.8 |
| RepLKNet-31B ● | $224^2$ | 79M | 15.4G | **295.5** | 83.5 |
| SLaK-B ● ● | $224^2$ | 95● / 122●M | 17.1● / 25.9●G | 245.4 | 84.0 |
| ConvNeXt-B-dcls ● | $224^2$ | 89M | 16.5G | 285.4 | **84.1** |

Table 2: **Classification accuracy on ImageNet-1K.** The inference throughput was calculated at inference using a single V100-32gb gpu and scaled to take into account all the optimizations used in Liu et al. (2022b). For the SLaK model, we report both the effective number of parameters and FLOPs returned by PyTorch ● and the one reported in Liu et al. (2022a) ●, that takes sparsity into account.

**Using ConvNeXt.** We present numerically in Table 2 and graphically in Fig. 1, the results obtained for ConvNeXt using the settings for ImageNet1k training with input crops of size $224 \times 224$, as described in Liu et al. (2022b): Table 5. Exponential Moving Average (EMA) (Polyak & Juditsky, 1992) was used, and for all models in Table 2, we report the accuracy found with this technique. We replaced ConvNeXt's depthwise separable convolutions (of kernel size $7 \times 7$), by 2D-DCLS ones of dilated kernel size $17 \times 17$ and of kernel count equal to 34 for the tiny variant, and 40 for the small and base variants. We used all the techniques previously described in Section 3 during training. From Table 2, we highlight the fact that ConvNeXt with DCLS convolutions always surpasses the ConvNeXt baseline in accuracy (gains ranging from 0.3 to 0.6) with the same number of parameters and only a little cost on throughput. We believe that this gain in accuracy is remarkable, given that we only replaced the depthwise convolutional layers, which represent just about 1% of the total number of parameters and 2% of the total number of FLOPs in ConvNeXt models. ConvNeXt model with a standard dilation of rate 2 performed poorly (see ConvNeXt-T-dil2). SLaK model performs about as well as DCLS but with a higher cost on throughput and parameters count.

DCLS can be seen as a kernel reparametrization technique that reduces the number of trainable parameters, and thus regularizes large kernels. For example, in the case of ConvNeXt-T-dcls, a $17 \times 17$ kernel (289 parameters) is parameterized by 34 triplets (x-position, y-position, weight), i.e. 102 parameters. The kernels that could be represented by the DCLS reparametrization constitute a subset of all possible dense kernels. In fact, by learning the suitable weights during training, a dense kernel could implement any DCLS one. It may therefore be counter-intuitive that DCLS leads to higher accuracy than a dense $17 \times 17$ convolution layer (see ConvNeXt-T-ker17). The problem with dense convolutional layers having large kernel sizes is that the number of trainable parameters is huge, which makes learning impractical. Finally, we observe that after training, the DCLS position density is higher around the center of the RF (see Appendix 10), suggesting that the central region is the most important one (yet we experienced that reducing the dilated kernel size to values $< 17$ degrades the accuracy, so the positions far from the center also matter). Conversely, DC samples the whole RF uniformly, which is most likely sub-optimal, which could explain its poor performance (see ConvNeXt-T-dil2).

### 4.3 Empirical Evaluation on Downstream and Robustness Tasks

We now report the results found for semantic segmentation on the ADE20K dataset (Zhou et al., 2019) and for object detection on the COCO dataset (Lin et al., 2014) using ConvNeXt-dcls backbones. Note that the *depthwise implcit gemm* algorithm was not used for those tasks as it led to throughput issues. In addition, we present the results found for robustness tasks consisting of directly testing (without further tuning) the previously obtained ConvNeXt-dcls backbones on the following robustness benchmarks: ImageNet-C/$\overline{\text{C}}$/A/R/Sketch (Hendrycks & Dietterich, 2019; Mintun et al., 2021; Hendrycks et al., 2021b;a; Wang et al., 2019).

**Semantic segmentation on ADE20k.** The results obtained in semantic segmentation show an improvement in performance by the ConvNeXt-dcls tiny and base backbones with equal number of parameters and FLOPs (Table 3). As in Liu et al. (2021) and Bao et al. (2021), we evaluated the mIoU with single scale testing and used the exact same configurations as in Liu et al. (2022b).

| backbone | input crop. | mIoU (ss) | # param. | FLOPs | throughput (image / s) |
|---|---|---|---|---|---|
| ConvNeXt-T ● | $512^2$ | 46.0 | 60M | 939G | 23.9 |
| SLaK-T ● | $512^2$ | **47.1** | 65M | 945G | — |
| ConvNeXt-T-dcls ● | $512^2$ | **47.1** | 60M | 950G | 21.1 |
| ConvNeXt-S ● | $512^2$ | **48.7** | 82M | 1027G | 22.1 |
| ConvNeXt-S-dcls ● | $512^2$ | 48.4 | 82M | 1045G | 19.5 |
| ConvNeXt-B ● | $512^2$ | 49.1 | 122M | 1170G | 21.7 |
| ConvNeXt-B-dcls ● | $512^2$ | **49.3** | 122M | 1193G | 18.6 |

Table 3: **ADE20K validation results** using UperNet (Xiao et al., 2018). We report mIoU results with single-scale testing. FLOPs are based on input sizes of (2048, 512). The inference throughput was calculated at inference using a single A100-80gb gpu and for input sizes of (3, 512, 512).

**Object detection and segmentation on COCO.** All ConvNeXt-dcls backbones have shown a noticeable improvement in average accuracy on both the object detection and segmentation tasks on the COCO dataset, again at iso-parameters and iso-FLOPS (Table 4). We only tested with Cascade Mask-RCNN (Cai & Vasconcelos, 2018) and used the exact same configurations as in Liu et al. (2022b).

| backbone | FLOPs | TPUT | $AP^{box}$ | $AP^{box}_{50}$ | $AP^{box}_{75}$ | $AP^{mask}$ | $AP^{mask}_{50}$ | $AP^{mask}_{75}$ |
|---|---|---|---|---|---|---|---|---|
| | | Cascade Mask-RCNN $3\times$ schedule | | | | | | |
| ResNet-50 | 739G | – | 46.3 | 64.3 | 50.5 | 40.1 | 61.7 | 43.4 |
| X101-32 | 819G | – | 48.1 | 66.5 | 52.4 | 41.6 | 63.9 | 45.2 |
| X101-64 | 972G | – | 48.3 | 66.4 | 52.3 | 41.7 | 64.0 | 45.1 |
| Swin-T | 745G | – | 50.4 | 69.2 | 54.7 | 43.7 | 66.6 | 47.3 |
| ConvNeXt-T ● | 741G | 11.6 | 50.4 | 69.1 | 54.8 | 43.7 | 66.5 | 47.3 |
| CNeXt-dcls-T ● | 751G | 11.2 | **51.2** | 69.9 | 55.7 | **44.5** | 67.5 | 48.3 |
| Swin-S | 838G | – | 51.9 | 70.7 | 56.3 | 45.0 | 68.2 | 48.8 |
| ConvNeXt-S ● | 827G | 11.2 | 51.9 | 70.8 | 56.5 | 45.0 | 68.4 | 49.1 |
| CNeXt-dcls-S ● | 844G | 10.5 | **52.8** | 71.6 | 57.6 | **45.6** | 69.0 | 49.3 |
| Swin-B | 982G | – | 51.9 | 70.5 | 56.4 | 45.0 | 68.1 | 48.9 |
| ConvNeXt-B ● | 964G | 11.1 | 52.7 | 71.3 | 57.2 | 45.6 | 68.9 | 49.5 |
| CNeXt-dcls-B ● | 987G | 10.3 | **53.0** | 71.5 | 57.7 | **46.0** | 69.3 | 50.0 |

Table 4: **COCO object detection and segmentation results** using Cascade Mask-RCNN. Average Precision of the ResNet-50 and X101 models are from (Liu et al., 2021). FLOPs are calculated with image size (3, 1280, 800). The inference throughput ("TPUT") was calculated at inference using a single A100-80gb gpu and for input sizes of (3, 512, 512).

**Robustness Evaluation on ImageNet-C/$\overline{\text{C}}$/A/R/Sketch.** ConvNeXt-dcls backbones show very good performances when it comes to robustness. This is illustrated by the results obtained for the different benchmarks we have tried and for which we have reconducted the experiments. All of them show a gain in classification accuracy with DCLS, except SK with the S model (Table 5).

| Model | FLOPs / Params | Clean | C($\downarrow$) | $\overline{\text{C}}$($\downarrow$) | A | R | SK |
|---|---|---|---|---|---|---|---|
| ResNet-50 | 4.1/25.6 | 76.1 | 76.7 | 57.7 | 0.0 | 36.1 | 24.1 |
| ConvNeXt-T ● | 4.5/28.6 | 82.1 | 41.6 | 41.2 | 23.5 | 47.6 | 33.8 |
| ConvNeXt-dcls-T ● | 5.0/28.6 | **82.5** | **41.5** | **39.7** | **23.9** | **47.8** | **34.7** |
| ConvNeXt-S ● | 8.7/50.2 | 83.1 | 38.9 | 37.8 | 30.1 | 50.1 | **37.1** |
| ConvNeXt-dcls-S ● | 9.5/50.2 | **83.7** | **37.8** | **35.2** | **33.7** | **50.4** | 36.7 |
| ConvNeXt-B ● | 15.4/88.6 | 83.8 | 37.0 | 35.7 | 35.5 | 51.7 | 38.2 |
| ConvNeXt-dcls-B ● | 16.5/88.6 | **84.1** | **36.3** | **34.3** | **36.8** | **52.6** | **38.4** |

Table 5: **Robustness evaluation of ConvNeXt-dcls**. We reconducted this study for ConvNeXt. For ImageNet-C and ImageNet-Cbar, the error is reported rather than the accuracy. It was calculated for both datasets by taking the average error over 5 levels of noise severity and over all the noise categories available in the datasets.

# 5 RELATED WORK

One of the studies that motivated the DCLS method is that of the *effective receptive field* (ERF) (Luo et al., 2016), which characterizes how much each input pixel in a receptive field can impact the output of a unit in the downstream convolutional layers of a neural network, leading to the notion of an effective receptive field. Among the findings of this study, the most relevant ones for ours are that not all pixels in a receptive field contribute equally to an output response, the kernel center has a much larger impact, and that the effective receptive field size increases linearly with the square root of convolutional layers. Given these findings, introducing a new degree of freedom by learning the positions of non-zero weights in dilated kernels might increase the expressive power of convolutional neural networks. A visual comparison between DCLS and non-DCLS ERFs is available in Appendix 12.

The work that relates the most to DCLS is that of deformable convolutions Dai et al. (2017) where offsets from the regular dilated grid are optimized. Deformable convolutions are used for several computer vision tasks such as object detection. Even if both approaches share the use of a rather similar bilinear interpolation, DCLS remains very different from deformable convolution in several aspects: firstly, deformable convolutions require the application of a regular convolution to get the offsets (which are thus input-dependent) that are then passed to the actual deformable method. Conversely, in DCLS method, a dilated kernel with learnable positions is constructed and then passed to the convolution operation, making the positions input-independent. Secondly, DCLS positions are channel-dependent, unlike deformable convolution ones. Thirdly, deformable convolutions have been developed for plain convolutions, not for depthwise separable ones. Finally, the number of extra learnable parameters in 2D deformable convolutions is the number of kernel elements in the preliminary convolution, precisely (channels_in // groups, $2 * K_H * K_W, K_H, K_W$) with $K_H$ and $K_W$ being the kernel height and width, which establishes a strong dependence between the offsets and the input feature map. Contrarily, in 2D-DCLS, the number of extra parameters dedicated to learning positions is simply twice the number of kernel weights (2, channels_out, channels_in // groups, $K_H, K_W$). We have not been able to compare our work to the deformable convolution v2 in ConvNeXt as the training becomes very slow (the throughput is divided by 4 for the ConvNeXt-tiny model). In addition, we noticed that the training loss grew at the first epochs, which means that the method is not adapted as a drop-in replacement of the depthwise separable convolution in the ConvNeXt architecture. Another input-dependent convolution method which seeks to learn the dilation rates rather than the kernel positions is the one of ADCNN (Yao et al., 2022). But in contrast to DCLS, this method uses regular grids with learnable rates.

DCLS is also similar to other input-independent kernel re-parameterization techniques, yet different from them. For example, in CKConv (Romero et al., 2021b) and FlexConv (Romero et al., 2021a), the kernel weights are not learned directly; what is learned is the continuous function that maps the positions to the weights. In Jacobsen et al. (2016), the kernel is modeled as a weighted sum of basis functions, which consist of centered Gaussian filters and their derivatives. Pintea et al. (2021) extended the approach by also learning the width of the Gaussians, which is equivalent to learning the optimal resolution. Shelhamer et al. (2019) proposed to factorize the kernel as the composition of a standard kernel with a structured Gaussian one. Finally, other methods such as(Worrall & Welling, 2019; Sosnovik et al., 2019; 2021a;b; Bekkers, 2019; Zhu et al., 2019), where the goal is to build a scale equivariant neural network, could be considered as similar to the approach. One limitation of all these studies is that only small datasets were used (e.g., CIFAR), and whether they scale well to larger and more challenging datasets like ImageNet1k is unknown. In addition, they cannot be used as a drop-in replacement for the depthwise separable convolution in ConvNeXt, at least in their current forms, so we could not benchmark them with DCLS.

## 6 CONCLUSION

In this work, we proposed DCLS, a new dilated convolution method where the positions of non-zero kernel elements are made learnable via backpropagation. The non-differentiability issue of learning discrete positions was circumvented by interpolation. We demonstrated that DCLS often outperforms the standard and the dilated convolution. We listed a number of techniques that improve the learning process of DCLS, in particular sharing the positions within stages was key. We provided evidence that searching for optimal positions of weights within a dilated kernel can improve not only the accuracy in image classification, but also in downstream and robustness tasks, using existing CNN architectures, without increasing their number of parameters. We reported a throughput overhead introduced by DCLS, but it remains marginal, provided that we use CNN architectures involving separable convolutions, which is the case for most modern CNNs, such as ConvNeXts. **Future work:** So far we have shown that DCLS can be used to drop-in replace standard convolution layers in existing architectures. We now would like to search for an architecture dedicated to DCLS, that would get the maximum benefit out of the method. In addition, we will explore alternatives to bilinear interpolation, e.g. bicubic. Finally, we wish to show that the method is also of interest in 1D and 3D use cases. For 1D cases, one can think of audio waveform generation, where dilated 1D convolution filters with large receptive fields are key, as, for example, in the autoregressive WaveNet model (Oord et al., 2016), and in generative adversarial networks (Yamamoto et al., 2020; Greshler et al., 2021). For 3D applications, video classification with architectures using 3D convolutions, e.g., X3D Feichtenhofer (2020), would be a good fit.

## ACKNOWLEDGMENTS

This work was performed using HPC resources from GENCI–IDRIS (Grant 2021-[AD011013219]) and from CALMIP (Grant 2021-[P21052]). Support from the ANR-3IA Artificial and Natural Intelligence Toulouse Institute is gratefully acknowledged. We would also like to thank the region of Toulouse Occitanie.

## REPRODUCIBILITY STATEMENT

The code of the method is based on PyTorch and available at `https://github.com/K-H-I smail/Dilated-Convolution-with-Learnable-Spacings-PyTorch`. Code and scripts to reproduce the training and the experiments are available at `https://github.com/K -H-Ismail/ConvNeXt-dcls`.

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

## 7 APPENDIX: PROOFS AND DERIVATION FOR THE DCLS METHOD

In the following, we show how to mathematically describe the DCLS kernel construction and how to explicitly calculate the gradients of the loss function with respect to weights and positions that are used in the backpropagation algorithm. These gradients are useful to implement the DCLS method in a way that is compatible with the automatic differentiation of PyTorch.

### 7.1 NOTATION AND PRELIMINARIES

We denote by $\lfloor \ \rfloor$ the floor function and we define its derivative by the zero function.

$$\forall x \in \mathbb{R}, \lfloor x \rfloor' \stackrel{\text{def}}{=} 0 \tag{6}$$

We denote by $m \in \mathbb{N}^*$ the number of kernel elements inside the constructed kernel and we refer to it as the "kernel count". Moreover, we denote respectively by $s_1, s_2 \in \mathbb{N}^* \times \mathbb{N}^*$, the sizes of the constructed kernel along the x-axis and the y-axis. The latter could be seen as the limits of the dilated kernel, and we refer to them as the "dilated kernel size".

The $n \times p$ matrix space over $\mathbb{R}$ is defined as the set of all $n \times p$ matrices over $\mathbb{R}$, and is denoted $\mathcal{M}_{n,p}(\mathbb{R})$.

The Frobenius inner product $\underset{\text{F}}{\times}$ of two matrices $\boldsymbol{A}$ and $\boldsymbol{B}$ of $\mathcal{M}_{n,p}(\mathbb{R})$ is defined by:

$$\boldsymbol{A} \underset{\text{F}}{\times} \boldsymbol{B} = \text{tr}(\boldsymbol{A}^T \boldsymbol{B})$$

Where "tr" stands for the trace of the square matrix $\boldsymbol{A}^T \boldsymbol{B}$.

The characters $w$, $p^1$ and $p^2$ respectively stand for the weight, the position of that weight along the x-axis (width) and its position along the y-axis (height) in the scalar case while the bold $\boldsymbol{w} = (w_i)_{1 \leq i \leq m}$, $\boldsymbol{p}^1 = (p_i^1)_{1 \leq i \leq m}$ and $\boldsymbol{p}^2 = (p_i^2)_{1 \leq i \leq m}$ respectively stand for the weight, the width-position of that weight and its height-position in the vector case.

The proofs and algorithms that will be shown in the next subsections are made for the case of tensors with one input channel and one output channel. Without loss of generality, those proofs and algorithms hold for the general case of 4D tensors and higher by considering and applying them channel-wise.

### 7.2 2D-DCLS, SCALAR WEIGHT CASE

We begin by the case of a kernel containing only one element.

The function $f$ that defines the kernel construction in the scalar weight case is as follows:

$$\begin{aligned} f \colon \mathbb{R} \times \mathbb{R} \times \mathbb{R} &\to \mathcal{M}_{s_1,s_2}(\mathbb{R}) \\ w, p^1, p^2 &\mapsto \boldsymbol{K} \end{aligned} \tag{7}$$

where $\forall i \in [\![ 1 \mathrel{..} s_1 ]\!], \forall j \in [\![ 1 \mathrel{..} s_2 ]\!]$:

$$\boldsymbol{K}_{ij} = \begin{cases} w\,(1 - r^1)\,(1 - r^2) & \text{if } i = \lfloor p^1 \rfloor,\ j = \lfloor p^2 \rfloor \\ w\,r^1\,(1 - r^2) & \text{if } i = \lfloor p^1 \rfloor + 1,\ j = \lfloor p^2 \rfloor \\ w\,(1 - r^1)\,r^2 & \text{if } i = \lfloor p^1 \rfloor,\ j = \lfloor p^2 \rfloor + 1 \\ w\,r^1\,r^2 & \text{if } i = \lfloor p^1 \rfloor + 1,\ j = \lfloor p^2 \rfloor + 1 \\ 0 & \text{otherwise} \end{cases} \tag{8}$$

and where the fractional parts are:

$$r^1 = \{p^1\} = p^1 - \lfloor p^1 \rfloor \quad \text{and} \quad r^2 = \{p^2\} = p^2 - \lfloor p^2 \rfloor \tag{9}$$

The constructed kernel $\boldsymbol{K}$ is zero except for at most the 4 adjacent positions that represent the 2D interpolation of the single weight $w$. Note that $\sum_{j=1}^{s_2} \sum_{i=1}^{s_1} \boldsymbol{K}_{ij} = w$.

We then define the scalar loss function as:

$$loss = g(f(w, p^1, p^2)) \tag{10}$$

with $g \colon \mathcal{M}_{s_1, s_2}(\mathbb{R}) \to \mathbb{R}$ a differentiable function that models the action of all the layers that will follow $f$ in the model.

By applying the chain rule we obtain:

$$\frac{\partial loss}{\partial w} = g'(f(w, p^1, p^2)) \underset{\text{F}}{\times} \frac{\partial f(w, p^1, p^2)}{\partial w} \tag{11}$$

$$\frac{\partial loss}{\partial p^1} = g'(f(w, p^1, p^2)) \underset{\text{F}}{\times} \frac{\partial f(w, p^1, p^2)}{\partial p^1} \tag{12}$$

$$\frac{\partial loss}{\partial p^2} = g'(f(w, p^1, p^2)) \underset{\text{F}}{\times} \frac{\partial f(w, p^1, p^2)}{\partial p^2} \tag{13}$$

with

$$g'(f(w, p^1, p^2)) = \frac{\partial loss}{\partial \boldsymbol{K}} = \frac{\partial loss}{\partial f(w, p^1, p^2)} \tag{14}$$

Let us put

$$g'(f(w, p^1, p^2)) = \boldsymbol{G} = \begin{bmatrix} g_{11} & g_{12} & \cdots & g_{1s_2} \\ g_{21} & \ddots & & g_{2s_2} \\ \vdots & & \ddots & \vdots \\ g_{s_1 1} & g_{s_1 2} & \cdots & g_{s_1 s_2} \end{bmatrix} \tag{15}$$

and let us consider two column vectors $\boldsymbol{x} = \begin{bmatrix} x_1 & x_2 & \cdots & x_{s_1} \end{bmatrix}^T$ of $\mathbb{R}^{s_1}$ and $\boldsymbol{y} = \begin{bmatrix} y_1 & y_2 & \cdots & y_{s_2} \end{bmatrix}^T$ of $\mathbb{R}^{s_2}$. We have:

$$\boldsymbol{x}^T f(w, p^1, p^2) \boldsymbol{y} = \sum_{i=1}^{s_1} \sum_{j=1}^{s_2} \boldsymbol{K}_{ij} x_i y_j \tag{16}$$

Since $\boldsymbol{K}$ is zero except for the 4 aforementioned positions, we have:

$$\begin{aligned} \boldsymbol{x}^T f(w, p^1, p^2) \boldsymbol{y} = &\; \boldsymbol{K}_{\lfloor p^1 \rfloor \lfloor p^2 \rfloor} \, x_{\lfloor p^1 \rfloor} \, y_{\lfloor p^2 \rfloor} \\ &+ \boldsymbol{K}_{\lfloor p^1 \rfloor + 1 \lfloor p^2 \rfloor} \, x_{\lfloor p^1 \rfloor + 1} \, y_{\lfloor p^2 \rfloor} \\ &+ \boldsymbol{K}_{\lfloor p^1 \rfloor \lfloor p^2 \rfloor + 1} \, x_{\lfloor p^1 \rfloor} \, y_{\lfloor p^2 \rfloor + 1} \\ &+ \boldsymbol{K}_{\lfloor p^1 \rfloor + 1 \lfloor p^2 \rfloor + 1} \, x_{\lfloor p^1 \rfloor + 1} \, y_{\lfloor p^2 \rfloor + 1} \end{aligned} \tag{17}$$

By deriving this expression with respect to $w$, $p^1$ and $p^2$ we obtain:

$$
\begin{aligned}
\frac{\partial(\boldsymbol{x}^T f(w,p^1,p^2)\boldsymbol{y})}{\partial w} = \ & \left(1-r^1\right)\left(1-r^2\right) x_{\lfloor p^1 \rfloor}\, y_{\lfloor p^2 \rfloor} \\
& + r^1\left(1-r^2\right) x_{\lfloor p^1 \rfloor+1}\, y_{\lfloor p^2 \rfloor} \\
& + \left(1-r^1\right) r^2\, x_{\lfloor p^1 \rfloor}\, y_{\lfloor p^2 \rfloor+1} \\
& + r^1\, r^2\, x_{\lfloor p^1 \rfloor+1}\, y_{\lfloor p^2 \rfloor+1}
\end{aligned}
\tag{18}
$$

$$
\begin{aligned}
\frac{\partial(\boldsymbol{x}^T f(w,p^1,p^2)\boldsymbol{y})}{\partial p^1} = \ & w\,\big[ -\left(1-r^2\right) x_{\lfloor p^1 \rfloor}\, y_{\lfloor p^2 \rfloor} \\
& + \left(1-r^2\right) x_{\lfloor p^1 \rfloor+1}\, y_{\lfloor p^2 \rfloor} \\
& - r^2\, x_{\lfloor p^1 \rfloor}\, y_{\lfloor p^2 \rfloor+1} \\
& + r^2\, x_{\lfloor p^1 \rfloor+1}\, y_{\lfloor p^2 \rfloor+1}\big]
\end{aligned}
\tag{19}
$$

$$
\begin{aligned}
\frac{\partial(\boldsymbol{x}^T f(w,p^1,p^2)\boldsymbol{y})}{\partial p^2} = \ & w\,\big[ -\left(1-r^1\right) x_{\lfloor p^1 \rfloor}\, y_{\lfloor p^2 \rfloor} \\
& - r^1\, x_{\lfloor p^1 \rfloor+1}\, y_{\lfloor p^2 \rfloor} \\
& + \left(1-r^1\right) x_{\lfloor p^1 \rfloor}\, y_{\lfloor p^2 \rfloor+1} \\
& + r^1\, x_{\lfloor p^1 \rfloor+1}\, y_{\lfloor p^2 \rfloor+1}\big]
\end{aligned}
\tag{20}
$$

Because of the linearity of the differentiation in the three previous equations, we could write:

$$
\frac{\partial(\boldsymbol{x}^T f(w,p^1,p^2)\boldsymbol{y})}{\partial w} = \boldsymbol{x}^T \frac{\partial f(w,p^1,p^2)}{\partial w}\boldsymbol{y} = \boldsymbol{x}^T \boldsymbol{G}_w \boldsymbol{y}
\tag{21}
$$

$$
\frac{\partial(\boldsymbol{x}^T f(w,p^1,p^2)\boldsymbol{y})}{\partial p^1} = \boldsymbol{x}^T \frac{\partial f(w,p^1,p^2)}{\partial p^1}\boldsymbol{y} = \boldsymbol{x}^T \boldsymbol{G}_{p^1} \boldsymbol{y}
\tag{22}
$$

$$
\frac{\partial(\boldsymbol{x}^T f(w,p^1,p^2)\boldsymbol{y})}{\partial p^2} = \boldsymbol{x}^T \frac{\partial f(w,p^1,p^2)}{\partial p^2}\boldsymbol{y} = \boldsymbol{x}^T \boldsymbol{G}_{p^2} \boldsymbol{y}
\tag{23}
$$

where $\boldsymbol{G}_w$, $\boldsymbol{G}_{p^1}$, $\boldsymbol{G}_{p^2}$, respectively stand for the $s_1$ by $s_2$ matrices described below and which have zeros everywhere except at the four positions of interpolation.

$\boldsymbol{G}_w =$

$$
\begin{array}{cc}
& \qquad\quad \lfloor p^2 \rfloor \qquad\quad \lfloor p^2 \rfloor+1 \\
\begin{pmatrix}
0 & \cdots & 0 & 0 & \cdots & 0 \\
\vdots & \ddots & 0 & 0 & \reflectbox{$\ddots$} & \vdots \\
0 & 0 & (1-r^1)(1-r^2) & r^2(1-r^1) & 0 & 0 \\
0 & 0 & r^1(1-r^2) & r^1 r^2 & 0 & 0 \\
\vdots & \reflectbox{$\ddots$} & 0 & 0 & \ddots & \vdots \\
0 & \cdots & 0 & 0 & \cdots & 0
\end{pmatrix}
&
\begin{array}{l} \\ \\ \lfloor p^1 \rfloor \\ \lfloor p^1 \rfloor+1 \\ \\ \end{array}
\end{array}
\tag{24}
$$

$\boldsymbol{G}_{p^1} =$

$$
\begin{array}{cc}
& \qquad\quad \lfloor p^2 \rfloor \qquad\quad \lfloor p^2 \rfloor+1 \\
\begin{pmatrix}
0 & \cdots & 0 & 0 & \cdots & 0 \\
\vdots & \ddots & 0 & 0 & \reflectbox{$\ddots$} & \vdots \\
0 & 0 & -w(1-r^2) & -wr^2 & 0 & 0 \\
0 & 0 & w(1-r^2) & wr^2 & 0 & 0 \\
\vdots & \reflectbox{$\ddots$} & 0 & 0 & \ddots & \vdots \\
0 & \cdots & 0 & 0 & \cdots & 0
\end{pmatrix}
&
\begin{array}{l} \\ \\ \lfloor p^1 \rfloor \\ \lfloor p^1 \rfloor+1 \\ \\ \end{array}
\end{array}
\tag{25}
$$

$$\boldsymbol{G}_{p^2} =$$

$$
\begin{array}{c}
\phantom{xx} \lfloor p^2 \rfloor \phantom{xxxx} \lfloor p^2 \rfloor + 1 \\
\begin{pmatrix}
0 & \cdots & 0 & 0 & \cdots & 0 \\
\vdots & \ddots & 0 & 0 & \iddots & \vdots \\
0 & 0 & -w(1-r^1) & w(1-r^1) & 0 & 0 \\
0 & 0 & -wr^1 & wr^1 & 0 & 0 \\
\vdots & \iddots & 0 & 0 & \ddots & \vdots \\
0 & \cdots & 0 & 0 & \cdots & 0
\end{pmatrix}
\begin{array}{l}
\\ \\ \lfloor p^1 \rfloor \\ \lfloor p^1 \rfloor + 1 \\ \\ \\
\end{array}
\end{array}
\tag{26}
$$

From the three equations (21), (22) and (23), we can identify

$$\frac{\partial f(w, p^1, p^2)}{\partial w} = \boldsymbol{G}_w \tag{27}$$

$$\frac{\partial f(w, p^1, p^2)}{\partial p^1} = \boldsymbol{G}_{p^1} \tag{28}$$

$$\frac{\partial f(w, p^1, p^2)}{\partial p^2} = \boldsymbol{G}_{p^2} \tag{29}$$

Finally, we have:

$$
\begin{aligned}
\frac{\partial loss}{\partial w} = \boldsymbol{G} \underset{\text{F}}{\times} \boldsymbol{G}_w = {} & (1-r^1)(1-r^2) g_{\lfloor p^1 \rfloor \lfloor p^2 \rfloor} \\
& + r^1(1-r^2) g_{\lfloor p^1 \rfloor + 1 \lfloor p^2 \rfloor} \\
& + (1-r^1) r^2 g_{\lfloor p^1 \rfloor \lfloor p^2 \rfloor + 1} \\
& + r^1 r^2 g_{\lfloor p^1 \rfloor + 1 \lfloor p^2 \rfloor + 1}
\end{aligned}
\tag{30}
$$

$$
\begin{aligned}
\frac{\partial loss}{\partial p^1} = \boldsymbol{G} \underset{\text{F}}{\times} \boldsymbol{G}_{p^1} = {} & w \big[ -(1-r^2) g_{\lfloor p^1 \rfloor \lfloor p^2 \rfloor} \\
& + (1-r^2) g_{\lfloor p^1 \rfloor + 1 \lfloor p^2 \rfloor} \\
& - r^2 g_{\lfloor p^1 \rfloor \lfloor p^2 \rfloor + 1} \\
& + r^2 g_{\lfloor p^1 \rfloor + 1 \lfloor p^2 \rfloor + 1} \big]
\end{aligned}
\tag{31}
$$

$$
\begin{aligned}
\frac{\partial loss}{\partial p^2} = \boldsymbol{G} \underset{\text{F}}{\times} \boldsymbol{G}_{p^2} = {} & w \big[ -(1-r^1) g_{\lfloor p^1 \rfloor \lfloor p^2 \rfloor} \\
& - r^1 g_{\lfloor p^1 \rfloor + 1 \lfloor p^2 \rfloor} \\
& + (1-r^1) g_{\lfloor p^1 \rfloor \lfloor p^2 \rfloor + 1} \\
& + r^1 g_{\lfloor p^1 \rfloor + 1 \lfloor p^2 \rfloor + 1} \big]
\end{aligned}
\tag{32}
$$

In the next subsection, we will see how this result can be generalized to the vector case.

### 7.3   2D-DCLS, GENERAL CASE

The general case is the one where the weights $\boldsymbol{w} = \begin{bmatrix} w_1 & w_2 & \cdots & w_m \end{bmatrix}^T$ and the positions $\boldsymbol{p}^1 = \begin{bmatrix} p_1^1 & p_2^1 & \cdots & p_m^1 \end{bmatrix}^T$, $\boldsymbol{p}^2 = \begin{bmatrix} p_1^2 & p_2^2 & \cdots & p_m^2 \end{bmatrix}^T$ are stored in vectors, with the fractional parts $\boldsymbol{r}^1 = \{\boldsymbol{p}^1\} = \boldsymbol{p}^1 - \lfloor \boldsymbol{p}^1 \rfloor = \begin{bmatrix} r_1^1 & r_2^1 & \cdots & r_m^1 \end{bmatrix}^T$ and $\boldsymbol{r}^2 = \{\boldsymbol{p}^2\} = \boldsymbol{p}^2 - \lfloor \boldsymbol{p}^2 \rfloor = \begin{bmatrix} r_1^2 & r_2^2 & \cdots & r_m^2 \end{bmatrix}^T$ extended as well.

The function $f$ defined in equation (7) is then extended to the function $F$ defined as follows:

$$
\begin{aligned}
F\colon \quad & \mathbb{R}^m \times \mathbb{R}^m \times \mathbb{R}^m \to \mathcal{M}_{s_1, s_2}(\mathbb{R}) \\
& \boldsymbol{w}, \boldsymbol{p}^1, \boldsymbol{p}^2 \mapsto \boldsymbol{K} = \sum_{i=1}^{m} f(w_i, p_i^1, p_i^2)
\end{aligned}
\tag{33}
$$

The constructed kernel $\boldsymbol{K}$ here is the result of a summation of the function $f$ defined in (7) over the elements of weight and position vectors. We then define the scalar loss function as in (10).

$$loss = g(F(\boldsymbol{w}, \boldsymbol{p}^1, \boldsymbol{p}^2)) \tag{34}$$

with $g \colon \mathcal{M}_{s_1, s_2}(\mathbb{R}) \to \mathbb{R}$ a differentiable function that models the action of all the layers that will follow $F$ in the model.

Let us put

$$g'(F(\boldsymbol{w}, \boldsymbol{p}^1, \boldsymbol{p}^2)) = \boldsymbol{G} = \begin{bmatrix} g_{11} & g_{12} & \cdots & g_{1s_2} \\ g_{21} & \ddots & & g_{2s_2} \\ \vdots & & \ddots & \vdots \\ g_{s_1 1} & g_{s_1 2} & \cdots & g_{s_1 s_2} \end{bmatrix} \tag{35}$$

As in (11), (12) and (13), by applying the chain rule we obtain:

$\forall i \in [\![1 .. m]\!]$:

$$\frac{\partial loss}{\partial w_i} = g'(F(\boldsymbol{w}, \boldsymbol{p}^1, \boldsymbol{p}^2)) \underset{\mathrm{F}}{\times} \frac{\partial F(\boldsymbol{w}, \boldsymbol{p}^1, \boldsymbol{p}^2)}{\partial w_i} \tag{36}$$

$$\frac{\partial loss}{\partial p_i^1} = g'(F(\boldsymbol{w}, \boldsymbol{p}^1, \boldsymbol{p}^2)) \underset{\mathrm{F}}{\times} \frac{\partial F(\boldsymbol{w}, \boldsymbol{p}^1, \boldsymbol{p}^2)}{\partial p_i^1} \tag{37}$$

$$\frac{\partial loss}{\partial p_i^2} = g'(F(\boldsymbol{w}, \boldsymbol{p}^1, \boldsymbol{p}^2)) \underset{\mathrm{F}}{\times} \frac{\partial F(\boldsymbol{w}, \boldsymbol{p}^1, \boldsymbol{p}^2)}{\partial p_i^2} \tag{38}$$

with

$$g'(F(\boldsymbol{w}, \boldsymbol{p}^1, \boldsymbol{p}^2)) = \frac{\partial loss}{\partial \boldsymbol{K}} = \frac{\partial loss}{\partial F(\boldsymbol{w}, \boldsymbol{p}^1, \boldsymbol{p}^2)} \tag{39}$$

Let us put this time

$$g'(F(\boldsymbol{w}, \boldsymbol{p}^1, \boldsymbol{p}^2)) = \boldsymbol{G} = \begin{bmatrix} g_{11} & g_{12} & \cdots & g_{1s_2} \\ g_{21} & \ddots & & g_{1s_2} \\ \vdots & & \ddots & \vdots \\ g_{s_1 1} & g_{12} & \cdots & g_{s_1 s_2} \end{bmatrix} \tag{40}$$

Using the definition of $F$, and by substituting the last equation in (36), (37) and (38), we have:

$$\frac{\partial loss}{\partial w_i} = \boldsymbol{G} \underset{\mathrm{F}}{\times} \frac{\partial \sum_{i=1}^m f(w_i, p_i^1, p_i^2)}{\partial w_i} \tag{41}$$

$$\frac{\partial loss}{\partial p_i^1} = \boldsymbol{G} \underset{\mathrm{F}}{\times} \frac{\partial \sum_{i=1}^m f(w_i, p_i^1, p_i^2)}{\partial p_i^1} \tag{42}$$

$$\frac{\partial loss}{\partial p_i^2} = \boldsymbol{G} \underset{\mathrm{F}}{\times} \frac{\partial \sum_{i=1}^m f(w_i, p_i^1, p_i^2)}{\partial p_i^2} \tag{43}$$

And we know that:

$\forall (i,j) \in [\![1 .. m]\!]^2$:

$$i \neq j \implies \frac{\partial f(w_j, p_j^1, p_j^2)}{\partial w_i} = \frac{\partial f(w_j, p_j^1, p_j^2)}{\partial p_i^1} = \frac{\partial f(w_j, p_j^1, p_j^2)}{\partial p_i^2} = 0$$

which simplifies equations (41), (42) and (43) to

$\forall i \in [\![1 .. m]\!]$ :

$$\frac{\partial loss}{\partial w_i} = \boldsymbol{G} \underset{\text{F}}{\times} \frac{\partial f(w_i, p_i^1, p_i^2)}{\partial w_i} \tag{44}$$

$$\frac{\partial loss}{\partial p_i^1} = \boldsymbol{G} \underset{\text{F}}{\times} \frac{\partial f(w_i, p_i^1, p_i^2)}{\partial p_i^1} \tag{45}$$

$$\frac{\partial loss}{\partial p_i^2} = \boldsymbol{G} \underset{\text{F}}{\times} \frac{\partial f(w_i, p_i^1, p_i^2)}{\partial p_i^2} \tag{46}$$

We can notice that we are brought back to the scalar case, and we deduce from (30), (31) and (32) the gradients of the loss function with respect to weights and positions in the general case:

$\forall i \in [\![1 .. m]\!]$ :

$$
\begin{aligned}
\left(\frac{\partial loss}{\partial \boldsymbol{w}}\right)_i = {} & (1 - r_i^1)\,(1 - r_i^2)\,g_{\lfloor p_i^1 \rfloor \lfloor p_i^2 \rfloor} \\
& + r_i^1\,(1 - r_i^2)\,g_{\lfloor p_i^1 \rfloor + 1 \lfloor p_i^2 \rfloor} \\
& + (1 - r_i^1)\,r_i^2\,g_{\lfloor p_i^1 \rfloor \lfloor p_i^2 \rfloor + 1} \\
& + r_i^1\,r_i^2\,g_{\lfloor p_i^1 \rfloor + 1 \lfloor p_i^2 \rfloor + 1}
\end{aligned}
\tag{47}
$$

$$
\begin{aligned}
\left(\frac{\partial loss}{\partial \boldsymbol{p}^1}\right)_i = {} & w_i\,\big[ -(1 - r_i^2)\,g_{\lfloor p_i^1 \rfloor \lfloor p_i^2 \rfloor} \\
& + (1 - r_i^2)\,g_{\lfloor p_i^1 \rfloor + 1 \lfloor p_i^2 \rfloor} \\
& - r_i^2\,g_{\lfloor p_i^1 \rfloor \lfloor p_i^2 \rfloor + 1} \\
& + r_i^2\,g_{\lfloor p_i^1 \rfloor + 1 \lfloor p_i^2 \rfloor + 1} \big]
\end{aligned}
\tag{48}
$$

$$
\begin{aligned}
\left(\frac{\partial loss}{\partial \boldsymbol{p}^2}\right)_i = {} & w_i\,\big[ -(1 - r_i^1)\,g_{\lfloor p_i^1 \rfloor \lfloor p_i^2 \rfloor} \\
& - r_i^1\,g_{\lfloor p_i^1 \rfloor + 1 \lfloor p_i^2 \rfloor} \\
& + (1 - r_i^1)\,g_{\lfloor p_i^1 \rfloor \lfloor p_i^2 \rfloor + 1} \\
& + r_i^1\,g_{\lfloor p_i^1 \rfloor + 1 \lfloor p_i^2 \rfloor + 1} \big]
\end{aligned}
\tag{49}
$$

The results found for the general case are nothing but a component-wise application of the result obtained in the scalar case. In addition, we show in Appendix 7.4, the extension to the 1D and 3D convolution cases.

## 7.4   1D-DCLS, 3D-DCLS

We denote respectively by $s_1, s_2, s_3 \in \mathbb{N}^* \times \mathbb{N}^* \times \mathbb{N}^*$, the sizes of the constructed kernel along the x-axis, y-axis and the z-axis. Moreover, the $n \times p \times q$ tensor space of third dimension is denoted $\mathbb{R}^{n \times p \times q}$.

The function $f$ defined in (7) could be adapted in order to construct a suitable kernel for the 1D convolution in the scalar weight case as follows:

$$
\begin{aligned}
f_{1D} \colon \mathbb{R} \times \mathbb{R} &\to \mathbb{R}^s \\
w, p &\mapsto \boldsymbol{k}
\end{aligned}
\tag{50}
$$

where $\forall i \in [\![1 .. s]\!]$ :

$$
\boldsymbol{k}_i = \begin{cases} w\,(1 - r) & \text{if } i = \lfloor p \rfloor \\ w\,r & \text{if } i = \lfloor p \rfloor + 1 \\ 0 & \text{else} \end{cases}
\tag{51}
$$

and where the fractional part is:

$$r = \{p\} = p - \lfloor p \rfloor \tag{52}$$

Following the same construction in Appendix 7.3 we can show that the gradients of the loss function with respect to weights and positions in the general 1D case are:

$\forall i \in [\![1 .. m]\!]$:

$$\left( \frac{\partial loss}{\partial \boldsymbol{w}} \right)_i = (1 - r_i) \, g_{\lfloor p_i \rfloor} + r_i \, g_{\lfloor p_i \rfloor + 1} \tag{53}$$

$$\left( \frac{\partial loss}{\partial \boldsymbol{p}} \right)_i = w_i \, (g_{\lfloor p_i \rfloor + 1} - g_{\lfloor p_i \rfloor}) \tag{54}$$

Furthermore, we define the function $f_{3D}$, the suitable kernel construction function in the 3D convolution case, as such:

$$\begin{aligned} f \colon \mathbb{R} \times \mathbb{R} \times \mathbb{R} \times \mathbb{R} &\to \mathbb{R}^{s_1 \times s_2 \times s_3} \\ w, p^1, p^2, p^3 &\mapsto \mathbf{K} \end{aligned} \tag{55}$$

where $\forall i \in [\![1 .. s_1]\!], \forall j \in [\![1 .. s_2]\!], \forall l \in [\![1 .. s_3]\!]$:

$$\mathbf{K}_{ijl} = \begin{cases} w \, (1 - r^1) \, (1 - r^2) \, (1 - r^3) & \text{if } i = \lfloor p^1 \rfloor, \, j = \lfloor p^2 \rfloor, \, l = \lfloor p^3 \rfloor \\ w \, r^1 \, (1 - r^2) \, (1 - r^3) & \text{if } i = \lfloor p^1 \rfloor + 1, \, j = \lfloor p^2 \rfloor, \, l = \lfloor p^3 \rfloor \\ w \, (1 - r^1) \, r^2 \, (1 - r^3) & \text{if } i = \lfloor p^1 \rfloor, \, j = \lfloor p^2 \rfloor + 1, \, l = \lfloor p^3 \rfloor \\ w \, r^1 \, r^2 \, (1 - r^3) & \text{if } i = \lfloor p^1 \rfloor + 1, \, j = \lfloor p^2 \rfloor + 1, \, l = \lfloor p^3 \rfloor \\ w \, (1 - r^1) \, (1 - r^2) \, r^3 & \text{if } i = \lfloor p^1 \rfloor, \, j = \lfloor p^2 \rfloor, \, l = \lfloor p^3 \rfloor + 1 \\ w \, r^1 \, (1 - r^2) \, r^3 & \text{if } i = \lfloor p^1 \rfloor + 1, \, j = \lfloor p^2 \rfloor, \, l = \lfloor p^3 \rfloor + 1 \\ w \, (1 - r^1) \, r^2 \, r^3 & \text{if } i = \lfloor p^1 \rfloor, \, j = \lfloor p^2 \rfloor + 1, \, l = \lfloor p^3 \rfloor + 1 \\ w \, r^1 \, r^2 \, r^3 & \text{if } i = \lfloor p^1 \rfloor + 1, \, j = \lfloor p^2 \rfloor + 1, \, l = \lfloor p^3 \rfloor + 1 \\ 0 & \text{else} \end{cases} \tag{56}$$

and where the fractional parts are:

$$\begin{aligned} r^1 &= \{p^1\} = p^1 - \lfloor p^1 \rfloor \\ r^2 &= \{p^2\} = p^2 - \lfloor p^2 \rfloor \\ r^3 &= \{p^3\} = p^3 - \lfloor p^3 \rfloor \end{aligned} \tag{57}$$

We can show that the gradients of the loss function with respect to weights and positions in the general 3D case are:

$\forall i \in [\![1 \mathbin{..} m]\!]:$

$$
\begin{aligned}
\left(\frac{\partial loss}{\partial \boldsymbol{w}}\right)_i = {}& (1 - r_i^1)\,(1 - r_i^2)\,(1 - r_i^3)\, g_{\lfloor p_i^1 \rfloor \lfloor p_i^2 \rfloor \lfloor p_i^3 \rfloor} \\
& + r_i^1\,(1 - r_i^2)\,(1 - r_i^3)\, g_{\lfloor p_i^1 \rfloor + 1 \lfloor p_i^2 \rfloor \lfloor p_i^3 \rfloor} \\
& + (1 - r_i^1)\, r_i^2\,(1 - r_i^3)\, g_{\lfloor p_i^1 \rfloor \lfloor p_i^2 \rfloor + 1 \lfloor p_i^3 \rfloor} \\
& + r_i^1\, r_i^2\,(1 - r_i^3)\, g_{\lfloor p_i^1 \rfloor + 1 \lfloor p_i^2 \rfloor + 1 \lfloor p_i^3 \rfloor} \\
& + (1 - r_i^1)\,(1 - r_i^2)\, r_i^3\, g_{\lfloor p_i^1 \rfloor \lfloor p_i^2 \rfloor \lfloor p_i^3 \rfloor + 1} \\
& + r_i^1\,(1 - r_i^2)\, r_i^3\, g_{\lfloor p_i^1 \rfloor + 1 \lfloor p_i^2 \rfloor \lfloor p_i^3 \rfloor + 1} \\
& + (1 - r_i^1)\, r_i^2\, r_i^3\, g_{\lfloor p_i^1 \rfloor \lfloor p_i^2 \rfloor + 1 \lfloor p_i^3 \rfloor + 1} \\
& + r_i^1\, r_i^2\, r_i^3\, g_{\lfloor p_i^1 \rfloor + 1 \lfloor p_i^2 \rfloor + 1 \lfloor p_i^3 \rfloor + 1}
\end{aligned}
\tag{58}
$$

$$
\begin{aligned}
\left(\frac{\partial loss}{\partial \boldsymbol{p}^1}\right)_i = {}& w_i \big[ - (1 - r_i^2)\,(1 - r_i^3)\, g_{\lfloor p_i^1 \rfloor \lfloor p_i^2 \rfloor \lfloor p_i^3 \rfloor} \\
& + (1 - r_i^2)\,(1 - r_i^3)\, g_{\lfloor p_i^1 \rfloor + 1 \lfloor p_i^2 \rfloor \lfloor p_i^3 \rfloor} \\
& - r_i^2\,(1 - r_i^3)\, g_{\lfloor p_i^1 \rfloor \lfloor p_i^2 \rfloor + 1 \lfloor p_i^3 \rfloor} \\
& + r_i^2\,(1 - r_i^3)\, g_{\lfloor p_i^1 \rfloor + 1 \lfloor p_i^2 \rfloor + 1 \lfloor p_i^3 \rfloor} \\
& - (1 - r_i^2)\, r_i^3\, g_{\lfloor p_i^1 \rfloor \lfloor p_i^2 \rfloor \lfloor p_i^3 \rfloor + 1} \\
& + (1 - r_i^2)\, r_i^3\, g_{\lfloor p_i^1 \rfloor + 1 \lfloor p_i^2 \rfloor \lfloor p_i^3 \rfloor + 1} \\
& - r_i^2\, r_i^3\, g_{\lfloor p_i^1 \rfloor \lfloor p_i^2 \rfloor + 1 \lfloor p_i^3 \rfloor + 1} \\
& + r_i^2\, r_i^3\, g_{\lfloor p_i^1 \rfloor + 1 \lfloor p_i^2 \rfloor + 1 \lfloor p_i^3 \rfloor + 1} \big]
\end{aligned}
\tag{59}
$$

$$
\begin{aligned}
\left(\frac{\partial loss}{\partial \boldsymbol{p}^2}\right)_i = {}& w_i \big[ - (1 - r_i^1)\,(1 - r_i^3)\, g_{\lfloor p_i^1 \rfloor \lfloor p_i^2 \rfloor \lfloor p_i^3 \rfloor} \\
& - r_i^1\,(1 - r_i^3)\, g_{\lfloor p_i^1 \rfloor + 1 \lfloor p_i^2 \rfloor \lfloor p_i^3 \rfloor} \\
& + (1 - r_i^1)\,(1 - r_i^3)\, g_{\lfloor p_i^1 \rfloor \lfloor p_i^2 \rfloor + 1 \lfloor p_i^3 \rfloor} \\
& + r_i^1\,(1 - r_i^3)\, g_{\lfloor p_i^1 \rfloor + 1 \lfloor p_i^2 \rfloor + 1 \lfloor p_i^3 \rfloor} \\
& - (1 - r_i^1)\, r_i^3\, g_{\lfloor p_i^1 \rfloor \lfloor p_i^2 \rfloor \lfloor p_i^3 \rfloor + 1} \\
& - r_i^1\, r_i^3\, g_{\lfloor p_i^1 \rfloor + 1 \lfloor p_i^2 \rfloor \lfloor p_i^3 \rfloor + 1} \\
& + (1 - r_i^1)\, r_i^3\, g_{\lfloor p_i^1 \rfloor \lfloor p_i^2 \rfloor + 1 \lfloor p_i^3 \rfloor + 1} \\
& + r_i^1\, r_i^3\, g_{\lfloor p_i^1 \rfloor + 1 \lfloor p_i^2 \rfloor + 1 \lfloor p_i^3 \rfloor + 1} \big]
\end{aligned}
\tag{60}
$$

$$
\begin{aligned}
\left(\frac{\partial loss}{\partial \boldsymbol{p}^3}\right)_i = {}& w_i \big[ - (1 - r_i^1)\,(1 - r_i^2)\, g_{\lfloor p_i^1 \rfloor \lfloor p_i^2 \rfloor \lfloor p_i^3 \rfloor} \\
& - r_i^1\,(1 - r_i^2)\, g_{\lfloor p_i^1 \rfloor + 1 \lfloor p_i^2 \rfloor \lfloor p_i^3 \rfloor} \\
& - (1 - r_i^1)\, r_i^2\, g_{\lfloor p_i^1 \rfloor \lfloor p_i^2 \rfloor + 1 \lfloor p_i^3 \rfloor} \\
& - r_i^1\, r_i^2\, g_{\lfloor p_i^1 \rfloor + 1 \lfloor p_i^2 \rfloor + 1 \lfloor p_i^3 \rfloor} \\
& + (1 - r_i^1)\,(1 - r_i^2)\, g_{\lfloor p_i^1 \rfloor \lfloor p_i^2 \rfloor \lfloor p_i^3 \rfloor + 1} \\
& + r_i^1\,(1 - r_i^2)\, g_{\lfloor p_i^1 \rfloor + 1 \lfloor p_i^2 \rfloor \lfloor p_i^3 \rfloor + 1} \\
& + (1 - r_i^1)\, r_i^2\, g_{\lfloor p_i^1 \rfloor \lfloor p_i^2 \rfloor + 1 \lfloor p_i^3 \rfloor + 1} \\
& + r_i^1\, r_i^2\, g_{\lfloor p_i^1 \rfloor + 1 \lfloor p_i^2 \rfloor + 1 \lfloor p_i^3 \rfloor + 1} \big]
\end{aligned}
\tag{61}
$$

# 8 APPENDIX: THE 2D-DCLS KERNEL CONSTRUCTION ALGORITHM

In the following, we describe with pseudocode the forward and backward passes for kernel construction used in 2D-DCLS. In practice, $\mathbf{W}$, $\mathbf{P}^1$ and $\mathbf{P}^2$ are 3-D tensors of size (`channels_out`, `channels_in // groups`, `K_count`), but the algorithms presented here are easily extended to this case by applying them channel-wise.

---

**Algorithm 1** 2D-DCLS kernel construction forward pass

---

**Input:** $\mathbf{W}, \mathbf{P}^1, \mathbf{P}^2$ : vectors of dimension $m$
**Output:** $\mathbf{K}$ : the constructed kernel, of size $(s_1 \times s_2)$
  1: $\mathbf{K} \leftarrow 0$
  2: $\boldsymbol{p}^1 \leftarrow \lfloor \mathbf{P}^1 \rfloor; \quad \boldsymbol{p}^2 \leftarrow \lfloor \mathbf{P}^2 \rfloor$
  3: $\mathbf{R}^1 \leftarrow \mathbf{P}^1 - \boldsymbol{p}^1; \quad \mathbf{R}^2 \leftarrow \mathbf{P}^2 - \boldsymbol{p}^2$
  4: save_for_backward $(\boldsymbol{p}^1, \boldsymbol{p}^2, \mathbf{R}^1, \mathbf{R}^2)$
  5: **for** $i = 0 \rightarrow m - 1$ **do**
  6: $\quad \mathbf{K}[\boldsymbol{p}_i^1, \boldsymbol{p}_i^2] \mathrel{+}= \mathbf{W}_i * (1 - \mathbf{R}_i^1) * (1 - \mathbf{R}_i^2)$
  7: $\quad \mathbf{K}[\boldsymbol{p}_i^1 + 1, \boldsymbol{p}_i^2] \mathrel{+}= \mathbf{W}_i * (\mathbf{R}_i^1) * (1 - \mathbf{R}_i^2)$
  8: $\quad \mathbf{K}[\boldsymbol{p}_i^1, \boldsymbol{p}_i^2 + 1] \mathrel{+}= \mathbf{W}_i * (1 - \mathbf{R}_i^1) * (\mathbf{R}_i^2)$
  9: $\quad \mathbf{K}[\boldsymbol{p}_i^1 + 1, \boldsymbol{p}_i^2 + 1] \mathrel{+}= \mathbf{W}_i * (\mathbf{R}_i^1) * (\mathbf{R}_i^2)$
 10: **end for**

---

**Algorithm 2** 2D-DCLS kernel construction backward pass

---

**Input:** $GradK = \frac{\partial Loss}{\partial K}$ : matrix of dimension $(s_1 \times s_2)$
**Output:** $\frac{\partial Loss}{\partial W}, \frac{\partial Loss}{\partial P^1}, \frac{\partial Loss}{\partial P^2}$ : vectors of dimension $m$
  1: $\frac{\partial Loss}{\partial W} \leftarrow 0, \frac{\partial Loss}{\partial P^1} \leftarrow 0, \frac{\partial Loss}{\partial P^2} \leftarrow 0$
  2: $\boldsymbol{p}^1, \boldsymbol{p}^2, \mathbf{R}^1, \mathbf{R}^2 \leftarrow$ load_saved ( )
  3: **for** $i = 0 \rightarrow m - 1$ **do**
  4:

$$\frac{\partial Loss}{\partial W}[i] \mathrel{+}= \frac{\partial Loss}{\partial K}[\boldsymbol{p}_i^1, \boldsymbol{p}_i^2] * (1 - \mathbf{R}_i^1) * (1 - \mathbf{R}_i^2) + \frac{\partial Loss}{\partial K}[\boldsymbol{p}_i^1 + 1, \boldsymbol{p}_i^2] * \mathbf{R}_i^1 * (1 - \mathbf{R}_i^2)$$
$$+ \frac{\partial Loss}{\partial K}[\boldsymbol{p}_i^1, \boldsymbol{p}_i^2 + 1] * (1 - \mathbf{R}_i^1) * \mathbf{R}_i^2 + \frac{\partial Loss}{\partial K}[\boldsymbol{p}_i^1 + 1, \boldsymbol{p}_i^2 + 1] * \mathbf{R}_i^1 * \mathbf{R}_i^2$$

  5:

$$\frac{\partial Loss}{\partial P^1}[i] \mathrel{+}= \mathbf{W}_i * [-\frac{\partial Loss}{\partial K}[\boldsymbol{p}_i^1, \boldsymbol{p}_i^2] * (1 - \mathbf{R}_i^2) + \frac{\partial Loss}{\partial K}[\boldsymbol{p}_i^1 + 1, \boldsymbol{p}_i^2] * (1 - \mathbf{R}_i^2)$$
$$- \frac{\partial Loss}{\partial K}[\boldsymbol{p}_i^1, \boldsymbol{p}_i^2 + 1] * \mathbf{R}_i^2 + \frac{\partial Loss}{\partial K}[\boldsymbol{p}_i^1 + 1, \boldsymbol{p}_i^2 + 1] * \mathbf{R}_i^2]$$

  6:

$$\frac{\partial Loss}{\partial P^2}[i] \mathrel{+}= \mathbf{W}_i * [-\frac{\partial Loss}{\partial K}[\boldsymbol{p}_i^1, \boldsymbol{p}_i^2] * (1 - \mathbf{R}_i^1) - \frac{\partial Loss}{\partial K}[\boldsymbol{p}_i^1 + 1, \boldsymbol{p}_i^2] * \mathbf{R}_i^1$$
$$+ \frac{\partial Loss}{\partial K}[\boldsymbol{p}_i^1, \boldsymbol{p}_i^2 + 1] * (1 - \mathbf{R}_i^1) + \frac{\partial Loss}{\partial K}[\boldsymbol{p}_i^1 + 1, \boldsymbol{p}_i^2 + 1] * \mathbf{R}_i^1]$$

  7: **end for**

---

The 'for' loops in the algorithms are fully parallelized using GPU threads. The 2D-DCLS convolution with kernel construction is then obtained by applying the classical 2D-convolution provided natively by PyTorch or any other method such as the *depthwise implicit gemm* convolution method Ding et al. (2022) using the constructed kernel.

When considering a concurrent execution of this pseudocode, the additions may result, in case of overlapping, in a replacement of the overlapped values instead of the desired accumulation. This problem can be addressed by using atomic addition operations.

# 9 APPENDIX: THE DCLS KERNEL CONSTRUCTION ALGORITHM IN NATIVE PYTORCH

In the following, we describe with PyTorch code the DCLS construction module for 1D, 2D and 3D versions. The DCLS convolution method is obtained by using the constructed kernel as a weight for the native torch.nn.Conv{1,2,3}d, or another convolution method such as *depthwise implicit gemm* Megvii (2020).

```python
class ConstructKernel1d(Module):
    def __init__(self, out_channels,
    in_channels, groups, kernel_count, dilated_kernel_size):
        super().__init__()
        self.out_channels = out_channels
        self.in_channels = in_channels
        self.groups = groups
        self.dilated_kernel_size = dilated_kernel_size
        self.kernel_count = kernel_count
        I = torch.arange(0, dilated_kernel_size[0])
        I = I.expand(out_channels,
        in_channels//groups, kernel_count,-1).permute(3,0,1,2)
        self.I = Parameter(I, requires_grad=False)

        self.lim = torch.zeros(1)
        self.lim[0] = dilated_kernel_size[0]
        self.lim = self.lim.expand(out_channels, in_channels//groups,
                                   kernel_count, -1).permute(3,0,1,2)
        self.lim = Parameter(self.lim, requires_grad=False)

    def forward(self, W, P):
        P = P + self.lim // 2
        Pr = P
        P = P.floor()
        R = (Pr - P).expand(self.dilated_kernel_size[0],-1,-1,-1,-1)
        R1 = R.select(2,0); P1 = P.select(0,0)
        cond1 = (self.I == P1)
        cond2 = (self.I == P1+1)
        W1 = torch.where(cond1, 1.0, 0.0)
        W2 = torch.where(cond2, 1.0, 0.0)

        K = W1 + R1 * (W2 - W1)
        K = W * K
        K = K.sum(3)
        K = K.permute(1,2,0)
        return K

class ConstructKernel2d(Module):
    def __init__(self, out_channels, in_channels, groups,
    kernel_count, dilated_kernel_size):
        super().__init__()
        self.out_channels = out_channels
        self.in_channels = in_channels
        self.groups = groups
        self.dilated_kernel_size = dilated_kernel_size
        self.kernel_count = kernel_count
```

```python
        J = torch.arange(0,
        dilated_kernel_size[0]).expand(dilated_kernel_size[1],-1)
        I = torch.arange(0,
        dilated_kernel_size[1]).expand(dilated_kernel_size[0],-1)
        I = I.expand(out_channels,
        in_channels//groups, kernel_count,-1,-1).permute(3,4,0,1,2)
        J = J.expand(out_channels,
        in_channels//groups, kernel_count,-1,-1).permute(4,3,0,1,2)

        self.I = Parameter(I, requires_grad=False)
        self.J = Parameter(J, requires_grad=False)
        self.lim = torch.zeros(2)
        self.lim[0] = dilated_kernel_size[0]
        self.lim[1] = dilated_kernel_size[1];
        self.lim = self.lim.expand(out_channels, in_channels//groups,
                                  kernel_count, -1).permute(3,0,1,2)
        self.lim = Parameter(self.lim, requires_grad=False)

    def forward(self, W, P):
        P = P + self.lim // 2
        Pr = P
        P = P.floor()
        R = (Pr - P).expand(self.dilated_kernel_size[0],
        self.dilated_kernel_size[1],-1,-1,-1,-1)
        R1 = R.select(2,0); P1 = P.select(0,0)
        R2 = R.select(2,1); P2 = P.select(0,1)
        R1R2 = R1*R2
        cond1 = (self.I == P1)
        cond2 = (self.J == P2)
        cond3 = (self.I == P1+1)
        cond4 = (self.J == P2+1)
        W1 = torch.where(cond1*cond2, 1.0, 0.0)
        W2 = torch.where(cond1*cond4, 1.0, 0.0)
        W3 = torch.where(cond3*cond2, 1.0, 0.0)
        W4 = torch.where(cond3*cond4, 1.0, 0.0)
        K = W1 + R1R2*(W1 - W2 - W3 + W4) + R1*(W3 - W1) + R2*(W2-W1)
        K = W * K
        K = K.sum(4)
        K = K.permute(2,3,0,1)
        return K

class ConstructKernel3d(Module):
    def __init__(self, out_channels, in_channels, groups,
    kernel_count, dilated_kernel_size):
        super().__init__()
        self.out_channels = out_channels
        self.in_channels = in_channels
        self.groups = groups
        self.dilated_kernel_size = dilated_kernel_size
        self.kernel_count = kernel_count
        L = torch.arange(0,
        dilated_kernel_size[0]).expand(dilated_kernel_size[1],
        dilated_kernel_size[2],-1)
        J = torch.arange(0,
        dilated_kernel_size[1]).expand(dilated_kernel_size[0],
        dilated_kernel_size[2],-1)
        I = torch.arange(0,
        dilated_kernel_size[2]).expand(dilated_kernel_size[0],
        dilated_kernel_size[1],-1)
```

```python
        L = L.expand(out_channels,
        in_channels//groups,kernel_count,-1,-1,-1).permute(5,3,4,0,1,2)
        I = I.expand(out_channels,
        in_channels//groups,kernel_count,-1,-1,-1).permute(3,4,5,0,1,2)
        J = J.expand(out_channels,
        in_channels//groups,kernel_count,-1,-1,-1).permute(3,5,4,0,1,2)
        self.L = Parameter(L, requires_grad=False)
        self.I = Parameter(I, requires_grad=False)
        self.J = Parameter(J, requires_grad=False)
        self.lim = torch.zeros(3)
        self.lim[0] = dilated_kernel_size[0]
        self.lim[1] = dilated_kernel_size[1]
        self.lim[2] = dilated_kernel_size[2]
        self.lim = self.lim.expand(out_channels, in_channels//groups,
                            kernel_count, -1).permute(3,0,1,2)
        self.lim = Parameter(self.lim, requires_grad=False)

    def forward(self, W, P):
        P = P + self.lim // 2
        Pr = P
        P = P.floor()
        R = (Pr - P).expand(self.dilated_kernel_size[0],
        self.dilated_kernel_size[1], self.dilated_kernel_size[2],-1,-1,-1,-1)
        R1 = R.select(3,0); P1 = P.select(0,0)
        R2 = R.select(3,1); P2 = P.select(0,1)
        R3 = R.select(3,2); P3 = P.select(0,2)

        cond1 = (self.L == P1)
        cond2 = (self.I == P2)
        cond3 = (self.J == P3)
        cond4 = (self.L == P1+1)
        cond5 = (self.I == P2+1)
        cond6 = (self.J == P3+1)
        W1 = torch.where(cond1*cond2*cond3, 1.0, 0.0)
        W2 = torch.where(cond4*cond2*cond3, 1.0, 0.0)
        W3 = torch.where(cond1*cond5*cond3, 1.0, 0.0)
        W4 = torch.where(cond4*cond5*cond3, 1.0, 0.0)
        W5 = torch.where(cond1*cond2*cond6, 1.0, 0.0)
        W6 = torch.where(cond4*cond2*cond6, 1.0, 0.0)
        W7 = torch.where(cond1*cond5*cond6, 1.0, 0.0)
        W8 = torch.where(cond4*cond5*cond6, 1.0, 0.0)
        # needs a better computing
        K  = W1 * (1 - R1) * (1 - R2) * (1 - R3)
        K += W2 * R1       * (1 - R2) * (1 - R3)
        K += W3 * (1 - R1) * R2       * (1 - R3)
        K += W4 * R1       * R2       * (1 - R3)
        K += W5 * (1 - R1) * (1 - R2) * R3
        K += W6 * R1       * (1 - R2) * R3
        K += W7 * (1 - R1) * R2       * R3
        K += W8 * R1       * R2       * R3
        K = W * K
        K = K.sum(5)
        K = K.permute(3,4,0,1,2)
        return K
```

## 10 APPENDIX: HISTOGRAMS OF POSITIONS

In the following, we show as histograms over training epochs, the distribution of the four kernel positions for the 2D-DCLS convolutions of the ConvNeXt-T-dcls model. Note that there is no agglutination or edge effect around the kernel limits, and that the distributions are relatively stable, with a higher concentration around the center of the kernel. Individual positions, however, are constantly moving; see animation at:

```
https://github.com/K-H-Ismail/Dilated-Convolution-with-Learnable
-Spacings-PyTorch/blob/main/figs/animation.gif.
```

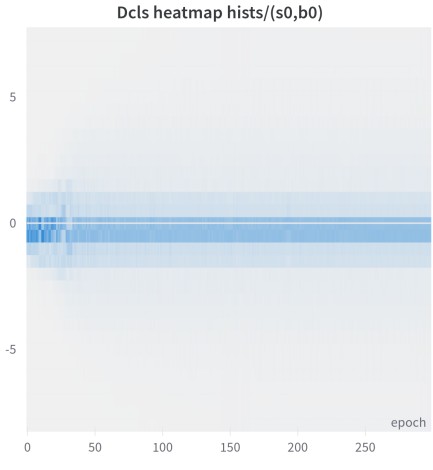

Figure 3: The distribution over epochs of kernel positions for the stage 0 of the ConvNeXt-T-dcls model.

Figure 4: Idem for stage 1.

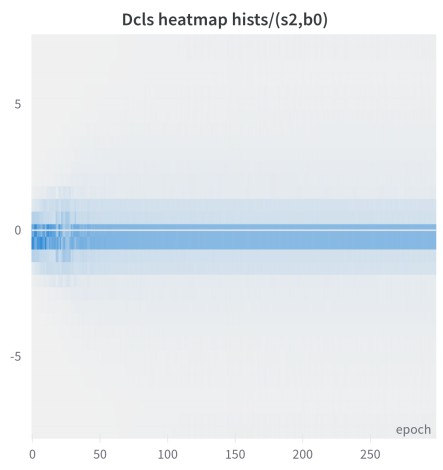

Figure 5: Idem for stage 2.

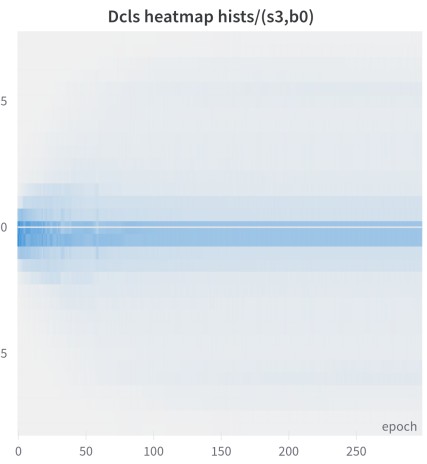

Figure 6: Idem for stage 3.

## 11 APPENDIX: SPEED CURVES AND LEARNING RATE SCHEDULE

Here, we plot the average speed curves of the four position tensors for the 2D-DCLS convolutions $V_\mathbf{P}$ of the ConvNeXt-T-dcls model as functions of the training epochs. The general formula for the

average speed of a DCLS position tensor of size $(cout, cin, m) \in \mathbb{N}^{*3}$, at epoch $t$, is as follows:

$$\forall t \in [\![1 .. t_{max}]\!]: \quad V_{\mathbf{P}}(t) = \frac{1}{cout \cdot cin \cdot m} \sum_{k=1}^{cout} \sum_{j=1}^{cin} \sum_{i=1}^{m} |P_{ijk}^t - P_{ijk}^{t-1}|$$

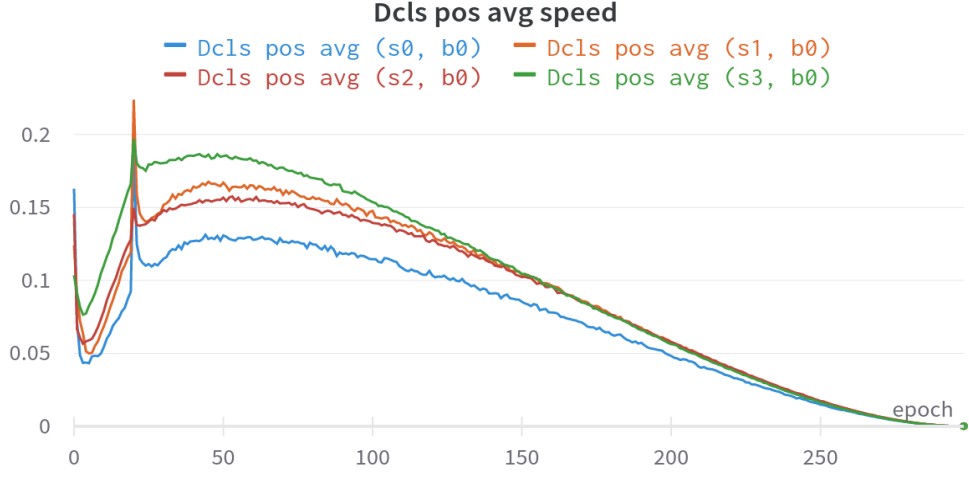

Figure 7: The average speed of the four position tensors for the ConvNeXt-T-dcls model as function of epochs.

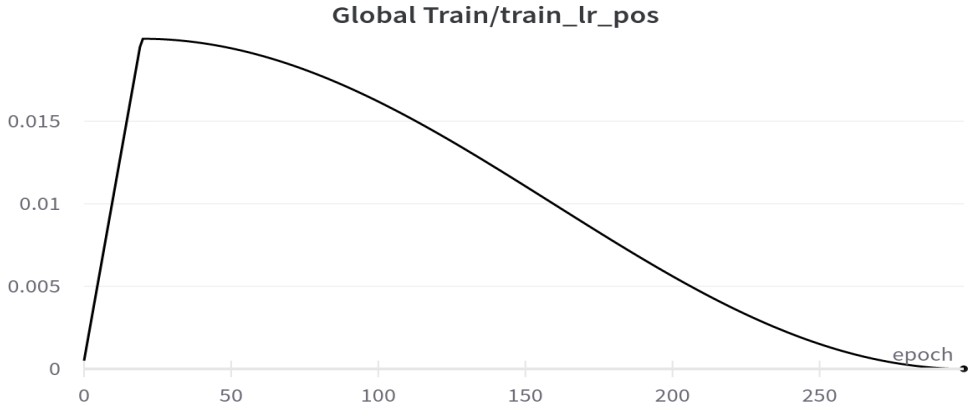

Figure 8: The learning schedule used for training ConvNeXt-T-dcls model.

At epoch 0, we can notice that the average speed is abnormally high, this is due to the fact that in the beginning, the positions are initialized randomly and we arbitrarily considered that $V_{\mathbf{P}}(0) = 0$, thus the large speed gap at initialization. Another peak can be seen at epoch 20, this one is due to the introduction of the repulsive loss (Thomas et al., 2019) at this precise epoch during training. This last causes a momentary increase in the average speed of the DCLS positions. In general, we can say that over epochs, the average DCLS positions follow the shape of the scheduler used in training.

## 12 APPENDIX: EFFECTIVE RECEPTIVE FIELDS COMPARISON

In the following, we show the effective receptive fields (ERF) calculated respectively for ConvNeXt-T-dcls, ConvNeXt-T with a standard dilated kernel of rate 2 and ConvNeXt-T models. The input crops used here are of size $1024 \times 1024$ and the heatmaps are normalized (between 0 and 1). These ERFs are obtained by finding (via backpropagation) the input pixels that most impact the activations of a pretrained model using samples of the dataset. We observe that the ERF of ConvNeXt-T-dcls has a particular shape that resembles a square with more prominent diagonals and medians. The ERF of ConvNeXt-T with a standard dilated kernel is larger but with gridding artifacts. In all plots, it seems that the center has more importance.

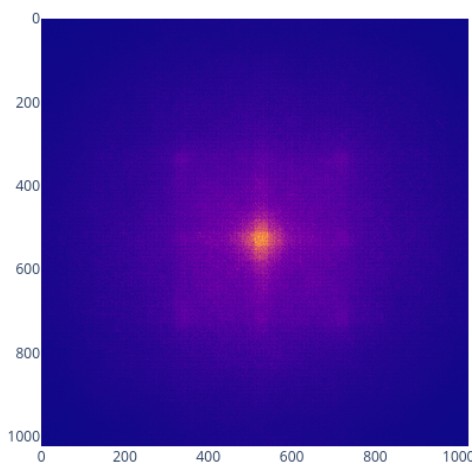

Figure 9: The effective receptive field (ERF) of the ConvNeXt-T-dcls model.

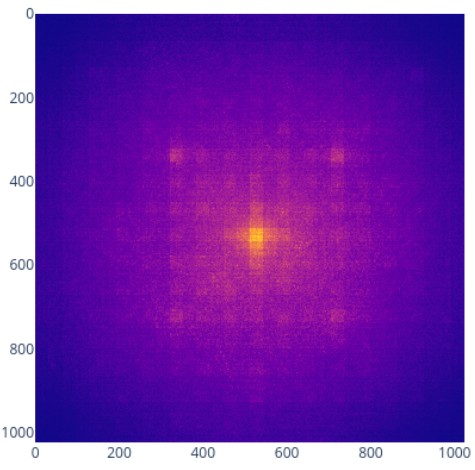

Figure 10: The effective receptive field of the ConvNeXt model with dilated kernels (dil rate 2) instead of the dense $7 \times 7$ ones.

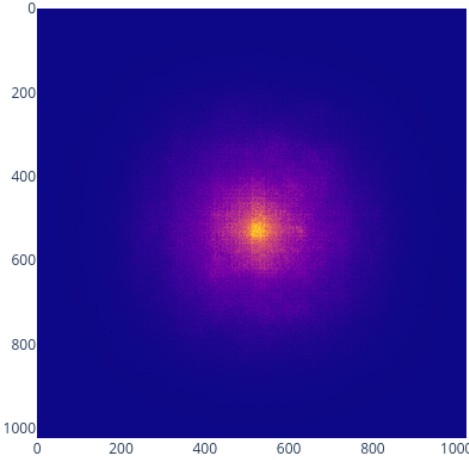

Figure 11: The effective receptive field of the ConvNeXt-T model.

## 13  APPENDIX: TIME MEASUREMENTS

We notice that when using the *depthwise implicit gemm* algorithm, DCLS forward is slightly slower than the PyTorch native Conv2d forward pass, but the DCLS backward pass is faster, in particular when increasing the model size.

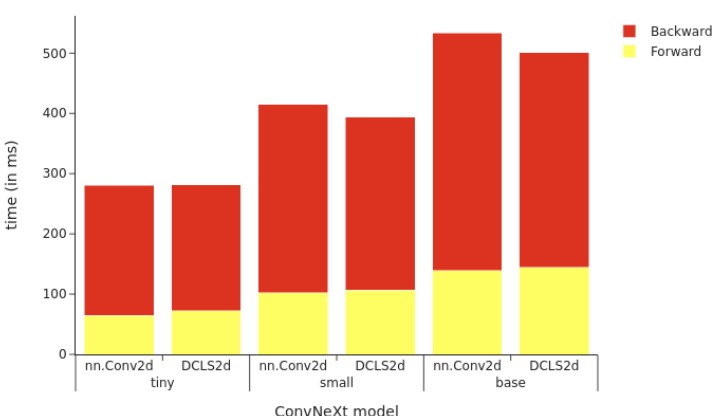

Figure 12: For the 3 ConvNeXt variants (tiny, small and base), we measure the elapsed time in ms for 1 forward + backward pass with a fixed batch size 128 and inputs of size (3,224,224) using DCLS2d convolution **accelerated by the *depthwise implicit* gemm algorithm**. Measures were carried using a single A100-80gb gpu. We also compare those timings to the 3 ConvNeXt baselines.

When not using the *depthwise implicit gemm* algorithm but the PyTorch native Conv2d in DCLS, the forward and backward are about twice slower.

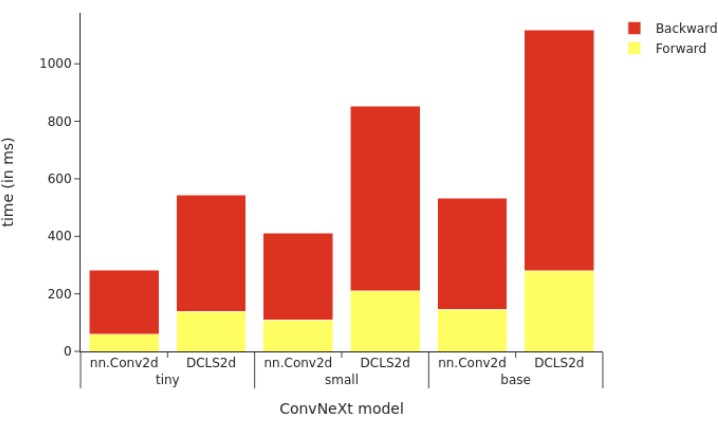

Figure 13: For the 3 ConvNeXt variants (tiny, small and base), we measure the elapsed time in ms for 1 forward + backward pass with a fixed batch size 128 and inputs of size (3,224,224) using DCLS2d convolution **with the PyTorch native 2D convolution algorithm**. Measures were carried using a single A100-80gb gpu. We compare those timings to the 3 ConvNeXt baselines. We also compare those timings to the 3 ConvNeXt baselines.

We notice that when using the *depthwise implicit gemm* algorithm, the DCLS construction algorithm is negligible in time compared to the convolution method for large input map sizes. For small map sizes (example 7x7), the DCLS construction algorithm takes as much time as the convolution.

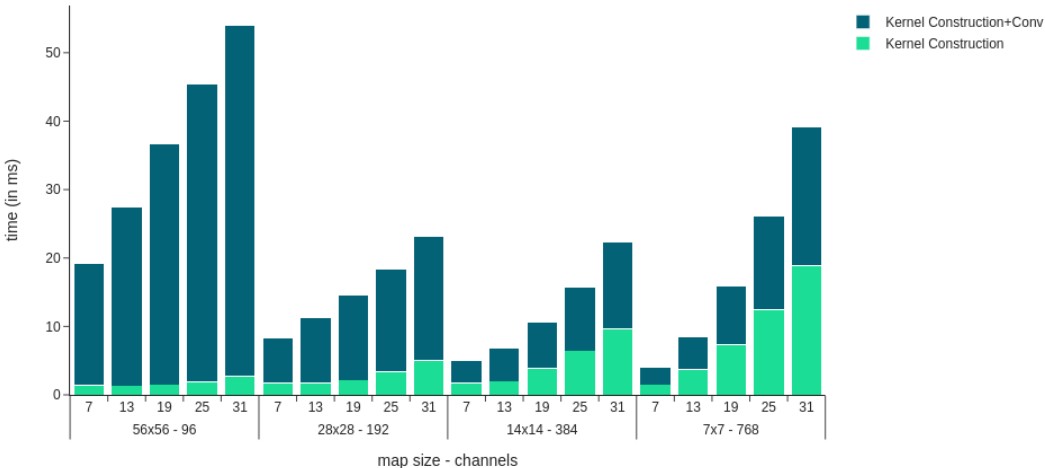

Figure 14: For different dilated kernel sizes (ranging from 7 to 31), and 4 different map sizes, we measure the elapsed time in ms for 1 forward + backward pass with a fixed batch size 128 and a fixed kernel count 34 using a single DCLS2d construct module with a 2D convolution **accelerated by the *depthwise implicit gemm* algorithm**. Measures were carried using a single Quadro RTX 8000 gpu.

When not using the *depthwise implicit gemm* algorithm but the PyTorch native Conv2d in DCLS, the construction algorithm time is significantly lower than the convolution time even for small input map sizes. This is due to the fact the convolution time with the PyTorch native Conv2d is significantly higher than the one with the *depthwise implicit gemm* algorithm.

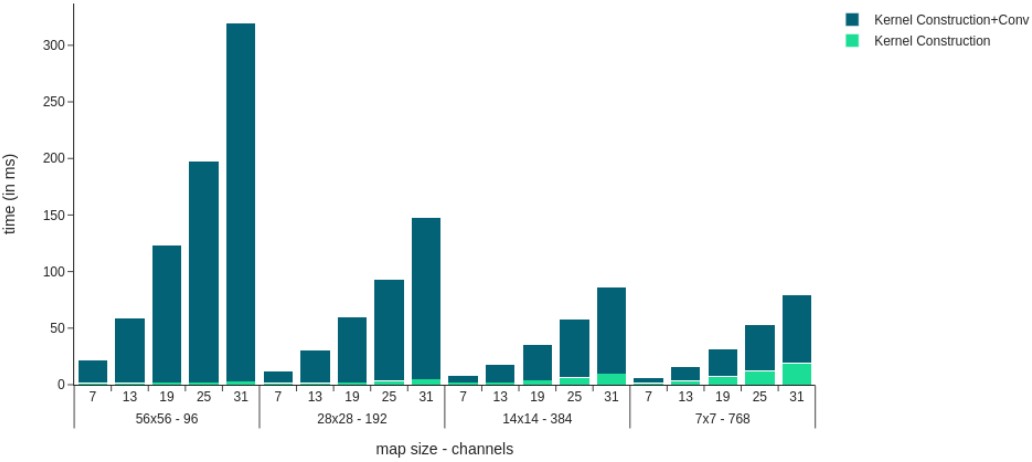

Figure 15: For different dilated kernel sizes (ranging from 7 to 31), and 4 different map sizes / channels, we measure the elapsed time in ms for 1 forward + backward pass with a fixed batch size 128 and a fixed kernel count 34 using a single DCLS2d construct module with **the PyTorch native 2D convolution algorithm**. Measures were carried using a single Quadro RTX 8000 gpu.

