# OpenReview forum: "Dilated convolution with learnable spacings"
_ICLR.cc/2023/Conference — ICLR 2023 poster_

### Official Review · Reviewer_2fMY · 2022-10-26

**Confidence:** 3
**Correctness:** 3
**Technical Novelty And Significance:** 3
**Empirical Novelty And Significance:** 3
**Recommendation:** 6

**Clarity, Quality, Novelty And Reproducibility:**

*clarity: as explained above, while the motivation and usefulness of such a method are straightforward, the model section could be significantly improved.
* novelty: I found at least [one reference](https://dl.acm.org/doi/abs/10.1016/j.patcog.2021.108369) not mentioned in the related work and that tries to address the same problem, so it may be that authors forgot to mention close work.
* reproducibility: the code provides direct alternatives to standard convolutional layers and can thus be used very easily.

**Strength And Weaknesses:**

Strengths:
* the dilation rate of dilated convolution layers requires a careful cross-validation, with sometimes very little intuition to guide the exploration, which becomes prohibitively expensive in networks that involve many dilated layers as they interact globally. Having 1) a way to replace cross-validation by backpropagation 2) handle irregular grids are very desirable features for such layers, to the potential impact is large.
* the method of authors is a drop-in replacement to standard convolutions and comes at a light increased computational cost, which makes it useful in practice and not just in an exploratory setting.
* the method is conceptually simple (reparametrization + bilinear interpolation) which is a clear plus
* section 3 is a great idea, it's rare to see a paper detail as much the various explorations of authors, including unsuccessful ones.

Weaknesses:
* the main weakness of the paper is that it completely ignores the groundbreaking advances that dilated convolutions brought to audio, where the scale of dependencies to handle is much higher than in images and where dilation rates are typically increase in an exponential fashion along the network to go up to receptive fields of 10^3 to 10^6 values. The most famous success of this approach is [Wavenet](https://arxiv.org/abs/1609.03499) which inspired many works in tasks such as [speech separation](https://arxiv.org/abs/1809.07454) or [audio coding](https://arxiv.org/abs/2107.03312). While I understand that extensive experiments in computer vision are already costly in time and resources, it is unfortunate that authors did not consider (or even mention) the huge opportunity that their method could represent for audio understanding and generation.
* the second big weakness is the model section is very austere and does not guide the reader at all. Not only notations are not detailed (one needs to go to appendix to get the definition of p1 and p2) but the reader is left with a matrix definition and two algorithms without further explanation or high-level description of the framework. I strongly recommend refactoring this section to make it more readable, as a high-level of the concepts, and pseudo-code if necessary belong more in the main text (in my opinion) than the exact backpropagation algorithm which takes half a page.
* as far as I understand, the authors use a standard convolution algorithm regardless of the level of sparsity of their kernel, which explains why they maintain a high computational cost that does not correlated with the reduced number of parameters. This is unlike standard dilated convolutions that exploit their regular grid to reshape inputs before applying the standard convolution with a kernel of size `k` rather than a kernel of size `s1*s2` (cf. this [tensorflow function](https://www.tensorflow.org/api_docs/python/tf/space_to_batch_nd)). It would be nice if authors discussed this issue.
* There is a lack of discussion. Do the authors consider the problem of learning the spacings now solved? If not, what would they like to improve in their method and how?

Questions:
* I assume the FLOPs reported for DCLS are at inference. It would be interesting to know how training time is affected by the forward and backward passes through the DCLS layers.
* As explained by authors, dilated convolutions allow increasing the receptive field with a limited parameter budget and computational cost. Recent approaches such as [CKConv](https://arxiv.org/abs/2102.02611) parametrize kernels of the size of the input with a small neural network, and compute this convolution in the Fourier domain. Such an approach offers both the limited parameter budget and computational cost without requiring dilation, and I would be interested to read (in the related work or as an answer) the author's view of such approaches.

Typos:
* "FlOPs" in Table 5 should be "FLOPs"

**Summary Of The Paper:**

Dilated or "à trous" convolutions allow increasing the receptive field of convolutional networks by orders of magnitude, at a constant time and space complexity. While a standard approach is to define the dilation rate as an hyperparameter, to define a regularly spaced grid, authors propose and adaptive alternative which, given a budget of kernel positions, learns where to place these positions on the grid.

They show improved or identical performance on computer vision tasks (classification, segmentation, detection), improved at constant parameter number and identical with fewer parameters.

**Summary Of The Review:**

Anyone who ever worked with dilated convolutions probably thought at some point "what if I could learn this pattern instead of cross-validating it?", and dilated convolutions are a fundamental building block of so many architectures that the potential impact of such a method is very significant. The main weakness of the paper is that they completely ignore the audio domain which is probably the field where dilated convolutions had the biggest impact. While this would be acceptable in computer vision conference, it's unfortunate in a machine learning conference to not consider such a field that would have been the perfect testbed for the author's method. In particular, the experimental setting chosen by the authors remains limited to very small kernels, which limits the understanding of the method's potential in more general settings.

---

> ### Author Response · Authors · 2022-11-16
> **Answer to reviewer 2fMy**
>
> The authors thank the reviewer 2fMy for his review, which helped to improve the quality of our manuscript. In the following, a response to the questions and remarks made by the reviewer:
>
> ### Weaknesses
>
> > The main weakness of the paper is that it completely ignores the groundbreaking advances that dilated convolutions brought to audio, where the scale of dependencies to handle is much higher than in images and where dilation rates are typically increase in an exponential fashion along the network to go up to receptive fields of $10^3$ to $10^6$ values.
>
>
> The reviewer is right, the method is even more promising for audio (we now briefly mention this in the Future work section).
> But we think that is a reason to accept the paper, not to reject it!
> As most reviewers pointed out, the amount of work we have done is already considerable. ICLR is a general ML conference, but that doesn’t mean vision-only papers cannot be accepted, and each year many vision-only papers are accepted. We now clearly say in the abstract of the paper that the focus of this paper is computer vision.
>
> > The second big weakness is the model section is very austere and does not guide the reader at all.
>
>
> Several reviewers pointed out that Section 2 (Kernel Construction in DCLS) was not very clear. This is due to the fact that the notations and preliminaries had been moved  to appendix 7 to save space. We understood that this might have affected the readability of the paper, so we moved the notations and preliminaries back in Section 2 for this new version, while refactoring this section to make it more understandable.
>
> > As far as I understand, the authors use a standard convolution algorithm regardless of the level of sparsity of their kernel.
>
> You are right, the current implementation does not leverage sparsity. This is now discussed at the bottom of page 4 in the item named "Dilated kernel size tuning". We also thought of adapting the im2col algorithm that is used by the standard, regular grid, dilated convolution. Unfortunately, it was not possible, since in DCLS the positions are different for each input channel and each output channel.
>
>
> > There is a lack of discussion. Do the authors consider the problem of learning the spacings now solved?
>
> The problem is solved in the sense that the solution we found works. But there may be ways to improve it.  In particular, we will explore alternatives
> to bilinear interpolation, e.g. bicubic. We now briefly mention this in the Future work section, but unfortunately, we do not have enough space for a long discussion.
>
> ### Questions
> > I assume the FLOPs reported for DCLS are at inference.
>
> Yes it is mentioned in the captions of the results tables.
>
> > It would be interesting to know how training time is affected by the forward and backward passes through the DCLS layers.
>
> In appendix 13, we added four bar plots in order to take this remark into consideration.
>
> > As explained by authors, dilated convolutions allow increasing the receptive field with a limited parameter budget and computational cost. Recent approaches such as CKConv parametrize kernels of the size of the input with a small neural network, and compute this convolution in the Fourier domain. Such an approach offers both the limited parameter budget and computational cost without requiring dilation, and I would be interested to read (in the related work or as an answer) the author's view of such approaches.
>
> Following the remarks of reviewers zKnb, 2NcJ and 2fMY , in section 5, we now cite all the related work suggested by the reviewers and compare their approaches to ours. In particular, we compare DCLS to the mentioned method.
>
> > Typos: "FlOPs" in Table 5 should be "FLOPs"
>
> Corrected, thanks again for your precision!
>
> PS : (18 Nov 22).
> > novelty: I found at least one reference not mentioned in the related work and that tries to address the same problem, so it may be that authors forgot to mention close work.
>
> We noticed the work of Jie Yaoa et al. "ADCNN: Towards learning adaptive dilation for convolutional neural networks" afterwards, and added it into Section 5 Related work before the deadline. Sorry for not doing it earlier.

---

### Official Review · Reviewer_N17u · 2022-10-27

**Confidence:** 4
**Correctness:** 3
**Technical Novelty And Significance:** 3
**Empirical Novelty And Significance:** 3
**Recommendation:** 8

**Clarity, Quality, Novelty And Reproducibility:**

I think the majority of the paper has very good clarity.

Pending my questions, I would place the quality and novelty pretty high, as I think it sits right in between several different important topics.

The reproducibility should also be high, as the code was provided.

**Strength And Weaknesses:**

Strength:
- The paper shows that the convolution coordinates in dilated convolution can be more flexible than a fixed, grid structure. In this sense, this work showcases the middle point between dilated convolution and deformable convolution (the latter has the added flexibility being input-dependent).
- Though this paper is framed as an extension to dilated convolution, in my opinion it really answers the broader question, which is whether or not the coordinates in convolution can be learned instead of being pre-determined. In this sense, it brings the concept of neural architecture search to a very elementary neuron level (i.e. convolution coordinates).
- The performance improvement is solid, considering that DCLS is building on competitive architectures (e.g. ConvNeXt) on competitive datasets / tasks (e.g. ImageNet classification).
- The majority of the paper is well-written and easy to follow, minus a few places which I will mention in the weaknesses below.

Weaknesses:
Q1 - The clarity of Section 2 can be greatly improved. $p^1$ and $p^2$ are not clearly defined as the x, y float location among $m$ convolution coordinates, and it is up to the reader to connect the dots. Also it wasn't until Section 3 when "three additional learnable parameters" is explicitly mentioned. Explicitly mentioning towards the beginning of Section 2 that "our extension entails that we learn the float coordinates ($p^1$, $p^2$ in the 2D case) for convolution input in addition to the true weight" would make things a lot easier to understand. Finally, the explanation here can be done in conjunction with Figure 2. Maybe add a big (e) panel which zooms in on four adjacent pixels and draw what $p^1$, $p^2$, $r^1$, $r^2$ mean.
Q2 - I understand that the length of Section 2 is limited. But I think one important question that needs to be answered in the main text is a clear intuition and explanation why the bilinear interpolation formulation unlocks the learning of the coordinates. How does the derivation change / where does the derivation break,  when $p$ must be integers or if $K$ only considers the closest element instead of the four neighbors? Empirically, will 'nearest' perform much worse than 'bilinear'? The conclusion section writes "circumvented by interpolation" but I did not find its justification in the main text.
Q3 - Why set a limit to the dilated kernel size? Ignoring all model capacity related concerns, does removing this limit always result in better performance?
Q4 - Deformable convolution was discussed in Section 5, which is very good because it is indeed very related. However, I think one angle the authors did not explicitly mention is the fact that deformable convolutions are *input-dependent*, and dilated convolution / DCLS are not. As such, the deformable convolution paper was able to show visualization where the kernel looks at different (but key) areas when presented with different inputs. Though DCLS also tries to be flexible in its convolution coordinates, I feel the intuition here may be less clear. I wonder if the authors have any comment on this, or if can show visualization of 10 randomly selected learned convolution input patterns?
Q5 - I appreciate the authors describing the important techniques needed to train DCLS. It is also mentioned in Section 6 that "in particular sharing the positions within stages was key". However, I do not see ablation study results / tables that support these claims. Having some of them, or at least the key ones, will strengthen the paper.
Q6 - Appendix 10, the text writes "the ERF of ConvNext-T-dcls is larger", but the ERF in Figure 9 seems smaller than that in Figure 11. Were the captions flipped?

Nit:
- Add space after "impractical." on page 7
- Cai & Vasconcelos (2018) should be \citep towards the end of page 7

**Summary Of The Paper:**

This paper proposes an advanced version of dilated convolution. In particular, this would mean learning the exact (float) spacing of the convolution locations in addition to the weights, instead of treating the former as a hyper-parameter. Learning of this spacing was enabled through bilinear interpolation as well as deriving the backpropagation rules. Empirically, the paper found that this additional degree of freedom can improve image recognition performance as well as downstream performance, even on very competitive CNN architectures such as ConvNeXt. The authors also provided important techniques that are needed to train this model successfully and analyses such as the empirical receptive field size.

**Summary Of The Review:**

Overall I feel positive about the paper at this stage. The problem is an interesting one; not addressed in an overly-complicated manner; and the experimental advantage is solid. But I also feel very strongly about my questions. I feel having concrete answers to those would make the paper even stronger.

=======================================

POST-REBUTTAL UPDATE

I have read the authors' rebuttal, which I think addresses my questions reasonably well. I also appreciate the authors' efforts in actively improving the paper throughout the process. I have increased my score which shows up as an 8, but that is because there is no 7 to choose between 6 and 8. I would really give this paper a "7 - Good paper, accept" rating, and personally vote for acceptance.

---

> ### Author Response · Authors · 2022-11-16
> **Answer to reviewer N17u - Part 1**
>
> The authors would like to sincerely thank the reviewer N17u for the high quality of his review. Furthermore, the authors greatly appreciate the fact that the reviewer understood the difference between DCLS and deformable convolution.
>
> > Q1 - The clarity of Section 2 can be greatly improved. $p^1$ and $p^2$ are not clearly defined as the x, y float location among $m$ convolution coordinates, and it is up to the reader to connect the dots. Also it wasn't until Section 3 when "three additional learnable parameters" is explicitly mentioned. Explicitly mentioning towards the beginning of Section 2 that "our extension entails that we learn the float coordinates ($p^1$, $p^2$ in the 2D case) for convolution input in addition to the true weight" would make things a lot easier to understand. Finally, the explanation here can be done in conjunction with Figure 2. Maybe add a big (e) panel which zooms in on four adjacent pixels and draw what $p^1$, $p^2$, $r^1$, $r^2$ mean.
>
> Several reviewers pointed out that Section 2 (Kernel Construction in DCLS) was not very clear. This is due to the fact that the notations and preliminaries had been moved  to appendix 7 to save space. We understood that this might have affected the readability of the paper, so we moved the notations and preliminaries back in Section 2 for this new version, while refactoring this section to make it more understandable. We have also added the sentence "our extension entails that we learn the float coordinate ..." specified by the reviewer.
>
> > Q2 - I understand that the length of Section 2 is limited. But I think one important question that needs to be answered in the main text is a clear intuition and explanation why the bilinear interpolation formulation unlocks the learning of the coordinates.
>
> Bilinear interpolation was for us the most "natural" choice when smoothing a problem with 2D integers. Other interpolation techniques such as cubic or bicubic interpolation can logically work in the same way or even give better results. Nevertheless, a 2D nearest neighbor interpolation would fail, because it is not differentiable.
>
> > How does the derivation change / where does the derivation break, when $p$ must be integers or if $K$ only considers the closest element instead of the four neighbors? Empirically, will 'nearest' perform much worse than 'bilinear'? The conclusion section writes "circumvented by interpolation" but I did not find its justification in the main text.
>
> When considering the derivation result in the general case of 2D DCLS for example, (these results are in appendix 7.3 in equations (47), (48) and (49)), the case in which all $p^1$ and $p^2$ must be integers is the one where $r^1$ and $r^2$ are equal to 0 as they are the fractional parts $(r^1 = p^1 - \lfloor p^1 \rfloor) $ and $(r^2 = p^2 - \lfloor p^2 \rfloor) $. By replacing $r^1$ and $r^2$ by 0 in equations (47), (48) and (49) it simplifies to :
>
> $ \forall i \in [ 1 \ .. \ m ] \colon$
>
> $(\frac{\partial loss}{\partial w})_ {i} = g _{ \lfloor p^{1} _i \rfloor  \lfloor p^{2} _i \rfloor} = g _{ p^{1} _i  p^{2} _i }$
>
> $(\frac{\partial loss}{\partial p^1})_ {i} = w _i (g _{ p^{1} _i  p^{2} _i } - g _{ p^{1} _i +1  p^{2} _i } ) $
>
> $(\frac{\partial loss}{\partial p^2})_ {i} = w _i (g _{ p^{1} _i  p^{2} _i + 1 } - g _{ p^{1} _i  p^{2} _i })$
>
> If we consider a nearest interpolation (at strictly more than 0.5 we decide that the position has switched to the next integer) instead, we should modify the definition of $r^1$ and $r^2$ to the following:
>
> $r^1 = H(p^1 - \lfloor p^1 \rfloor - 0.5) $
>  and
> $r^2 = H(p^2 - \lfloor p^2 \rfloor - 0.5) $
>
> with $H$ being the Heaviside step function:
>
> $\forall x \in \mathbb{R} \colon H(x) = 1 \text{ if }  x > 0 \text{ else } 0$.
>
> Everything else stays the same. This theoretically wouldn't work, as the derivative of the Heaviside step function is zero almost everywhere.
>
> > Q3 - Why set a limit to the dilated kernel size? Ignoring all model capacity related concerns, does removing this limit always result in better performance?
>
> Ignoring all additional cost on computational time, removing the dilated kernel size limit does not lead to better accuracy. We added this sentence in the paper: "For simplicity,
> we used the same dilated kernel size in all the model layers (7 for ResNet-50-dcls and 17 for
> ConvNeXt-dcls; larger values did not bring any gain in accuracy).". Therefore the limit 17x17 for the ConvNext-dcls model was found to be empirically optimal as we tested larger values (19x19, 21x21 ...) and that didn't improve training loss in the first part of training as our criterion was training loss in the first epochs (early stopping to save computation time).

---

> > ### Author Response · Authors · 2022-11-16
> > **Answer to reviewer N17u - Part 2**
> >
> > > Q4 - Deformable convolution was discussed in Section 5, which is very good because it is indeed very related. However, I think one angle the authors did not explicitly mention is the fact that deformable convolutions are input-dependent, and dilated convolution / DCLS are not.
> >
> > Following this remark, we emphasized the fact that DCLS is input-independent in contrast with deformable convolution which is input-dependent in Section 5 (Related work) paragraph 2.
> >
> >
> > > As such, the deformable convolution paper was able to show visualization where the kernel looks at different (but key) areas when presented with different inputs. Though DCLS also tries to be flexible in its convolution coordinates, I feel the intuition here may be less clear. I wonder if the authors have any comment on this, or if can show visualization of 10 randomly selected learned convolution input patterns?
> >
> > From what we understood, the reviewer would like a visualization of the learned kernel patterns. This was available in the Wandb link in appendix 10. We invite the reviewer to take a look at the animation following the link provided there. If it is not this visualisation that is being requested, we then politely ask the review to clarify what he means by "learned convolution input patterns".
> >
> > > Q5 - I appreciate the authors describing the important techniques needed to train DCLS. It is also mentioned in Section 6 that "in particular sharing the positions within stages was key". However, I do not see ablation study results / tables that support these claims. Having some of them, or at least the key ones, will strengthen the paper.
> >
> > We performed an ablation study on this particular technique (Synchronizing positions) and added this sentence in Section 3 : " (an ablation of this technique led to a 0.13\% accuracy drop on ImageNet1k with ConvNeXt-T-dcls)". We have not been able to perform an extensive ablation study (running the ablation for 300 epochs for every technique and perform it at least for 3 different seeds) as it will cost more gpu time than what we have at our disposal. Instead we validated those thechniques by performing early stopping and looking at the training loss as said before.
> >
> >
> > > Q6 - Appendix 10, the text writes "the ERF of ConvNext-T-dcls is larger", but the ERF in Figure 9 seems smaller than that in Figure 11. Were the captions flipped?
> >
> > The caption was not flipped, for the authors, the ERF in Figure 9 seems to be bigger, as it is a  subjective statement, we prefer to remove this sentence from the paper.
> >
> > > Nit: Add space after "impractical." on page 7.
> > Cai \& Vasconcelos (2018) should be citep towards the end of page 7
> >
> > Those have been corrected, thanks again for your precision !

---

> > > ### Comment · Reviewer_N17u · 2022-12-02
> > > **Appreciate the rebuttal**
> > >
> > > I appreciate the rebuttal and have increased my score from 6 to 7. See the updated part in "Summary Of The Review" section for more. Good luck!

---

### Official Review · Reviewer_8tHP · 2022-10-27

**Confidence:** 5
**Correctness:** 3
**Technical Novelty And Significance:** 2
**Empirical Novelty And Significance:** 2
**Recommendation:** 6

**Clarity, Quality, Novelty And Reproducibility:**

The quality of the paper itself as an artifact is high. The writing is good, the experiments well done, the baselines mostly appropriate (see deformable convolutions).

The work appears reasonably original, though the idea itself is very similar to Deformable Convolutional Networks (https://arxiv.org/abs/1703.06211) which have been widely used in the object detection community for awhile. The authors do spend a large portion of their related work contrasting the methods, and they are different, but not very different.

The paper has available code that presumably can be run to reproduce the results. I did not verify myself.

**Strength And Weaknesses:**

Strengths:

- The paper itself is well structured, well written, and very easy to follow.
- The experiments are thorough and well done, and include most of the basic data I would request as a reviewer (e.g. they have throughput, though they probably should also have it for object detection and semantic segmentation, and have appropriate baselines with modified receptive fields).
- The method is relatively simple to understand, and the paper correctly focuses on one single idea throughout, holding everything else constant. I didn't have to guess where their gains came from.

Main Weaknesses:

- The paper, and associated project, feel poorly motivated to me. I don't get the why of this project from several angles: Why do we want or need to learn the kernel locations? When do we expect this to help? And why would people use this?
  - The intro states `Our own investigation on ResNet and ConvNeXt with standard dilated convolution (Section 4.2) will lead to a similar conclusion. The failure of this method for classification tasks could be attributed to the great rigidity imposed by its regular grid as discussed in Wang & Ji (2018).`
    - I don't buy this motivation. The main use case of dilated convolutions was to grow the receptive field of models when applied to large images, like in semantic segmentation or other dense prediction tasks. For smaller images, like those used in classification like the authors do the dilated convolution study on, it makes no sense, as the receptive field is already the whole image.
- While the authors spend a lot of their related work contrasting the methods, it seems the situations (if any) you may want to use DCLS you may instead use deformable convolutions (https://arxiv.org/abs/1703.06211) which are popular in the object detection community. The authors should have compared directly to deformable convolutions.
- The results just aren't that strong. e.g. the authors state: `From Table 2, we highlight the fact that ConvNeXt with DCLS convolutions always surpasses the ConvNeXt baseline in accuracy (gains ranging from 0.3 to 0.6) with the same number of parameters and only a little cost on throughput`
  - I unfortunately don’t see these gains as meaningful at all. 0.3 - 0.6 accuracy improvement (e.g. 82.1 -> 82.5) on ImageNet is really small at the cost of ~7% lower throughput, but even more importantly, this would require researchers/users to use non-standard convolution operations (including compiling and packing custom CUDA operations with their models).
  - Furthermore, using non-standard convolution operations limits the platforms you can run this model on efficiently, and prohibits the advantages of different optimizations people may want to use (e.g. more efficient conv kernels for inference).


Other feedback:
- `“The problem with dense convolutional layers having large kernel sizes is that the number of trainable parameters is huge, which makes learning impractical.Finally, we observe that after training, the DCLS position density is higher around the center of the RF (see Appendix 8), suggesting that the central region is the most important one.”`
  - I don’t think the first claim here is really supported. The ConvNeXt-T-ker17 seems to do reasonably well. Perhaps it just needs a bit more regularization (which you can think of DCLS as a version of).
  - To the second point, the DCLS position density being higher around the center of the RF to me points to the fact that you probably don’t need that large of kernels.
- Re: Semantic Segmentation / Object Detection Results
  - Thank you for including these! These two tasks are a much better application of dilated convolutions than ImageNet classification. That said, in my opinion, we end up at the same place that the results, though a small improvement, are not really meaningful enough to justify using this method.
  - Throughput is also dropped here from the tables, and I suspect the small gains in accuracy (e.g. AP increases of 0.3-0.9 or mIoU increases/decreases of -0.3-1.1) are overshadowed by the increase in actual runtime.
- Re: Robustness: Is there a hypothesis driving this set of experiments? Why would these kernels lead to more robust networks? The gains here are also relatively minor.



**Summary Of The Paper:**

The paper proposes a technique to learn the shape of a dilated convolution kernel to allow via backprop the learning of the positions of the non-zero elements of the kernel. They do this by reparameterizing it as a series of weights (the “kernel count”), and a kernel window size (the “dilated kernel size”), and use bilinear interpolation to perform the convolution. They show modest gains on ImageNet image classification, Mscoco Object Detection, and ADE20k Semantic segmentation, at the cost of modestly decreased throughput.


**Summary Of The Review:**

While the paper itself as an artifact feels high quality, the experiments done are (mostly) appropriate and well executed, and the project requiring significant impressive engineering work, I just don't see this paper as impactful or meaningful. The results simply don't justify the extra complexity (in method, in code, in platform lock in) or run time hit. As a result, I don't see it as being something that gets built upon, used by others, etc.

Beyond the empirical and practical tradeoffs, the paper/project lacked a convincing underlying hypothesis of why we would want to do this at all. As a result, from reading the paper, the main takeaway of a reader will be "this specific idea, when implemented this specific way, works reasonably". There aren't generalizable insights that other researchers can build upon, learn from, or discuss.

My score is a result of a combination of these things. Because the paper feels pretty polished and finished, but I don't find the motivation or results themselves convincing, I see myself unlikely to raise my score.

**Update**: After conversations with other reviewers, I've tried to re-calibrate what I think the bar for acceptance should be. As a result, I've updated my score to a 6. After reading the rebuttal, I stand by my assessment that the actual paper quality (in writing, experiment design, code availability, etc) is high, but still have concerns about how well motivated or impactful the work will be. As a result, I am OK to see this accepted.

---

> ### Author Response · Authors · 2022-11-17
> **Answer to reviwer 8tHP - Part1**
>
> In the following, a response to the questions and remarks made by the reviewer:
>
>
> > The paper, and associated project, feel poorly motivated to me. I don't get the why of this project from several angles: Why do we want or need to learn the kernel locations? When do we expect this to help? And why would people use this?
>
>
> Firstly, the authors had a motivation: as explained in the introduction, recent CNN literature indicates that larger RFs are needed, even for ImageNet classification. Our method allows to increase RF without increasing the number of parameters, just like dilated convolutions, but in a more flexible way. Secondly, In Section 5 (Related Work), the authors explicitly state that :"One of the studies that **motivated** the DCLS method is that of the effective receptive field (ERF) (Luo et al., 2016), which characterizes how much each input pixel in a receptive field can impact the output of a unit in the downstream convolutional layers of a neural network, leading to the notion of an effective receptive field.".
> Besides, the authors wanted to explore the possibility that the fixed grid imposed by default by the standard dilated convolution is something to improve, and they had a good intuition in view of the empirical results obtained afterwards!
> Finally, we argue that a motivation is not strictly needed for something that empirically works!
>
>
> > I don't buy this motivation. The main use case of dilated convolutions was to grow the receptive field of models when applied to large images, like in semantic segmentation or other dense prediction tasks. For **smaller** images, like **those used in classification** like the authors do the dilated convolution study on, it makes no sense, as **the receptive field is already the whole image**.
>
> The image crops we use in ImageNet1k classification are not that small:  224x224.
> The ConvNeXt baseline uses 7x7 kernels, so the receptive field sizes in ConvNeXt-T blocks are:
>
> [$4^2$, $28^2$, $52^2$, $76^2$, $80^2$, $128^2$, $176^2$, $224^2$, $232^2$, $328^2$, $424^2$, $520^2$, $616^2$, $712^2$, $808^2$, $904^2$, $1000^2$, $1096^2$, $1112^2$, $1304^2$, $1496^2$, $1688^2$].
>
>
> So the RF encompasses the whole image only from the 7th block (last block of the 2nd stage).
> In addition, RepLKNet and SLaK show that increasing kernel sizes, up to respectively 31x31 and 51x51 helps on ImageNet1k  as well.
>
>
> > While the authors spend a lot of their related work contrasting the methods, it seems the situations (if any) you may want to use DCLS you may instead use deformable convolutions (https://arxiv.org/abs/1703.06211) which are popular in the object detection community. The authors should have compared directly to deformable convolutions.
>
> Thank you for your suggestion. We have tried deformable convolutions as a drop-in replacement of the 7x7 depthwise convolutions in ConvNeXt-T. As we report, in the manuscript, it did not work. This suggests that deformable convolutions are not suitable as a drop-in replacement of every depthwise separable convolution for the ConvNeXt model, at least in their current form.
>
>
> > The results just aren't that strong. e.g. the authors state: From Table 2, we highlight the fact that ConvNeXt with DCLS convolutions always surpasses the ConvNeXt baseline in accuracy (gains ranging from 0.3 to 0.6) with the same number of parameters and only a little cost on throughput.
>
>
> The gains in accuracy are worth the cost in throughput, as Fig. 1 demonstrates.
>
> For example, when moving from ConvNeXt-S to ConvNeXt-L:
>
> - Accuracy gain = 0.7\%
> - Parameter factor = 1.8
> - FLOP factor = 1.8
> - Throughput factor = 0.65
>
> When moving from ConvNeXt-S to ConvNeXt-S-DCLS:
>
> - Accuracy gain = 0.6\%
> - Parameter factor = 1.0
> - FLOP factor = 1.1
> - Throughput factor = 0.97
>
> DCLS is thus very appealing!
> We invite the reviewer, if it was not done before, to take a look at the Figure 1. (accuracy / latency figure, we recall that latency (s / img) is the inverse of throughput) to realize that DCLS is a very good trade off.
>
> The reviewer should keep in mind that, as acknowledged by the other reviewers, ImageNet1k is a very challenging and saturated dataset, and ConvNeXt is already a very optimized model.

---

> > ### Author Response · Authors · 2022-11-17
> > **Answer to reviwer 8tHP - Part2**
> >
> >
> > > but even more importantly, this would require researchers/users to use non-standard convolution operations (including compiling and packing custom CUDA operations with their models).
> >
> > We now provide 2 equivalent versions of DCLS, one using CUDA-PyTorch that requires a simple compilation, and another one using only native PyTorch modules that requires no compilation. The real PyTorch code of the kernel construction has been added in appendix 9 for this new version of the paper, and in the anonymous Github repository as well.
> >
> > > I don’t think the first claim here is really supported. The ConvNeXt-T-ker17 seems to do reasonably well. Perhaps it just needs a bit more regularization (which you can think of DCLS as a version of).
> >
> > You are totally right, DCLS can be seen as a regularization method (we have added this point in the manuscript, thank you), and an efficient one: ConvNeXt-T-ker17 reaches an accuracy of 82.0\% vs 82.5\% for ConvNeXt-T-DCLS, which is not negligible.
> >
> > > To the second point, the DCLS position density being higher around the center of the RF to me points to the fact that you probably don’t need that large of kernels.
> >
> > We experienced that reducing the dilated kernel size to values < 17 degrades the accuracy, so the positions far from the center also matter. We have added this point in the manuscript.
> >
> > > Thank you for including these! These two tasks are a much better application of dilated convolutions than ImageNet classification. That said, in my opinion, we end up at the same place that the results, though a small improvement, are not really meaningful enough to justify using this method.
> >
> > We also thought that the best use-cases for our method would be semantic segmentation and object detection with large images, rather than ImageNet classification, but empirically, we found the opposite! Nevertheless, DCLS has shown an improvement in object detection and semantic segmentation.
> >
> > > Throughput is also dropped here from the tables, and I suspect the small gains in accuracy (e.g. AP increases of 0.3-0.9 or mIoU increases/decreases of -0.3-1.1) are overshadowed by the increase in actual runtime.
> >
> > The throughput has been added to those tables in the new version.
> >
> > > Re: Robustness: Is there a hypothesis driving this set of experiments? Why would these kernels lead to more robust networks? The gains here are also relatively minor.
> >
> > We already explained our motivation, and we argue that the gains are not minor.
> >
> >
> > > The work appears reasonably original, though the idea itself is very similar to Deformable Convolutional Networks (https://arxiv.org/abs/1703.06211) which have been widely used in the object detection community for awhile. The authors do spend a large portion of their related work contrasting the methods, and they are different, but not very different.
> >
> >
> > We all agree that our method is different from previous proposals, and in particular from Deformable convolution. The statement “they are different, but not very different” is a subjective statement.
> >
> >
> >
> > > Because the paper feels pretty polished and finished, but I don't find the motivation or results themselves convincing, **I see myself unlikely to raise my score**.
> >
> > We hope that the reviewer will change his mind. As said above, the gains in accuracy on ImageNet1k are worth the increase in throughput (Fig. 1). Drop-in replacing a standard depthwise conv by a DCLS one is very easy, so we expect that most of the CNN community will do it after the publication this paper, especially for ImageNet classification.

---

> > > ### Comment · Reviewer_8tHP · 2022-12-02
> > > **Response to Authors**
> > >
> > > First, sorry for the extremely late reply. I appreciate the thoroughness of the rebuttal and several of the points have helped my understanding. I unfortunately don't have time to respond to each comment, but I want to note I updated my score to a 6 in the main review and updated my reasoning.
> > >
> > > I wanted to quickly respond to a few more small points in the rebuttal.
> > >
> > > `Finally, we argue that a motivation is not strictly needed for something that empirically works!`
> > >
> > > First, I better understand the motivation after other reviewers helpfully describe how this fits into the overall literature space, so I don't think this paper goes to this extreme, but I really want to challenge this idea. I hope the community can move past writing papers that change some small architectural detail in an existing network and publish slightly better numbers than a baseline on ImageNet/Coco/etc.
> > >
> > > Unless these tweaks are well motivated and no-brainer wins, I find these papers add very little to the scientific community.
> > >
> > > `Re: Pytorch code vs cuda`
> > > It'd be nice if you shared some rough throughput numbers of the custom cuda kernels vs the native pytorch approach. If the numbers in this paper are from the cuda kernels, but the paper claims "our new convolution is just some simple pytorch code" that would be incredibly misleading.

---

> > > > ### Author Response · Authors · 2022-12-09
> > > > **About the native PyTorch code vs. the CUDA extension**
> > > >
> > > > The authors thank the reviewer 8tHP for his response.
> > > >
> > > > To answer the question about the native PyTorch code vs. the extension written in CUDA, here is the throughput comparison for the 3 backbones presented in the paper for the classification task:
> > > >
> > > >
> > > > | Model       | Version     | Throughput  |
> > > > | ----------- | ----------- | ----------- |
> > > > | ConvNeXt-dcls-T      | cuda ext        | 725.3      |
> > > > | ConvNeXt-dcls-T      | torch    | 692.0      |
> > > > | ConvNeXt-dcls-S      | cuda ext        |  433.4      |
> > > > | ConvNeXt-dcls-S      | torch      |  417.7       |
> > > > | ConvNeXt-dcls-B      | cuda ext        |  285.4      |
> > > > | ConvNeXt-dcls-B      | torch     | 281.2      |
> > > >
> > > >
> > > > We notice that the overhead of using DCLS written with native modules is small (less than 5%) and decreases with the model size. This was to be expected as we already know that most of the time elapsed in the DCLS method does not come from the construction of the kernel (whether it is done in native PyTorch or with the CUDA extension) but rather from the subsequent convolution applied on this constructed kernel (see appendix. 13: Time Measurements).
> > > > Note that these throughput (as well as those presented in the paper, were obtained using a single V100 GPU - 32 gb and averaged over 100 runs of which 20 are warm-up runs). Moreover, the CUDA extension is a low-level rewrite of the native PyTorch code that can probably be further improved to become slightly faster.

---

### Official Review · Reviewer_2NcJ · 2022-10-27

**Confidence:** 3
**Correctness:** 4
**Technical Novelty And Significance:** 2
**Empirical Novelty And Significance:** 3
**Recommendation:** 5

**Clarity, Quality, Novelty And Reproducibility:**

Apart from the mathematical description of the bilinear interpolation, the paper is clearly written.
The quality is good, mostly from the experimental point of view.
The work is not particularly novel, but it's experimental insights are valuable.
Their code is made available.

**Details Of Ethics Concerns:**

No concerns

**Strength And Weaknesses:**

**Strengths**
1. The paper proposes a relatively simple, but effect idea
2. The method is extensively tested on 2D image tasks and results are clearly presented.
3. The method has practical value in that it seems to consistently improve performance at only a moderate reduction of througput. Moreover, the authors published their code with the paper.
4. The paper is transparent in also reporting negative findings; it reports things that worked and things tried, but didn't work

**Weaknesses**
1. The presentation of the bilinear interpolation is really confusing and possibly incorrect.
    * Equation 1 defines the kernel in terms of a function f but the function f is nowhere explicitly defined. The definition is somehow implicit in equation 3.
    * p^1 and p^2 are not defined. I suppose these are the "x" and "y" coordinates (centers) of the pixels. In the start of section 2, $n$ is not defined, nor is $p$. This is in some sense OK because the labels are only used to indicated the # rows and columns in a matrix, but it confuses with the notation of p for the positions.
    * In equation 1 however, index i is used to iterate over the weights/centers of the kernel, wherease in (3) it is used to iterate over the rows of K. This is somewhat confusing.
    * Even more so as in (3) the p^1 and p^j no longer have indices, which p are are talking about?
    * After a while I figured out that indeed this is just bilinear interpolation where each of the rows in (3) correspond to each of the 4 nearest corner points, but I feel like the presentation could have been much more effective.
2. Recent works on learnable receptive field sizes seem to be missing. In particlar the works of Romero et al. which show the benefit of (extremely) large receptive field sizes:
    * Romero, D. W., Kuzina, A., Bekkers, E. J., Tomczak, J. M., & Hoogendoorn, M. (2021). Ckconv: Continuous kernel convolution for sequential data. arXiv preprint arXiv:2102.02611.
and learnable receptive field sizes:
    * Romero, D. W., Bruintjes, R. J., Tomczak, J. M., Bekkers, E. J., Hoogendoorn, M., & van Gemert, J. C. (2021). Flexconv: Continuous kernel convolutions with differentiable kernel sizes. arXiv preprint arXiv:2110.08059.

    The idea of using kernels parametrized by a set of shifted weights has been done before by Dai et al. (already discussed in the paper) by shifted dirac deltas, as in this paper, or shifted Gaussians as in:
    * Jacobsen, J. H., Van Gemert, J., Lou, Z., & Smeulders, A. W. (2016). Structured receptive fields in cnns. In Proceedings of the IEEE Conference on Computer Vision and Pattern Recognition (pp. 2610-2619).
    * Shelhamer, E., Wang, D., & Darrell, T. (2019). Blurring the line between structure and learning to optimize and adapt receptive fields. arXiv preprint arXiv:1904.11487.
    * Pintea, S. L., Tömen, N., Goes, S. F., Loog, M., & van Gemert, J. C. (2021). Resolution learning in deep convolutional networks using scale-space theory. IEEE Transactions on Image Processing, 30, 8342-8353.

    The idea of sparse localized basis functions with learnable centers has also already been extended to group convolutional neural networks in
    * Bekkers, E. J. (2019, September). B-Spline CNNs on Lie groups. In International Conference on Learning Representations.
3. It should be discussed what the current work has to add to the above works. I believe that the emperical study presented in this paper is valuable in itself, but I believe some of the obtained insights have also already been discussed in the above works.

Minor details:
* It would be nice to give a bit more detail on what depthwise implicit gemm convolution entails. It is mentioned several times and a high level description could help the reader without them having to look up the original paper.
* Regarding the position initialization (section 3). "In an attempt to facilitate learning, we chose an initial distribution close to the one obtained at the end of training", is this reasonable? What if the initial distribution was chosen poorly, then also the one at the end of training would be right? "Yet in practice, ..." so would you recommend using uniform initialization then instead?
* On "Dilated kernel size tuning" I can imagine that the overhead could be independent from the constrained kernel size. Since convolution and kernel construction are linear, one could do the interpolation on the input feature maps instead (for the m different shifts) and then simply combine the results with a linear layer afterwards.
* In related work the effective RF size is discussed and the reader is referred to appendix 10. The idea of looking at the ERFs is interesting, however, app 10 does not give much detail on how the "heatmaps" are obtained.
* When discussin the method in comparison to deformable convolutions, it mentions "Firstly, in deformable convolutions, the offsets ..." I do not consider this to be a difference. Parametrizing the locations as offsets rather then absolute points (which are offsets to an origin) boils down to the same thing. Also, the "secondly", it indeed describes a difference, but I consider dilated convs to be strictly more general than the proposed work (i.e. one could decide with deformable convs not to make the kernels dependent on the input)

**Summary Of The Paper:**

The paper describes a method for learning sparse convolution kernels (like dilated convolutions) that consists of a weighted sum of shifted pixels. I.e. the kernels can be understood to be expanded of a basis of shifted dirac-deltas, and the learnable parameters are the shifts for each "pixel" and its corresponding weight. The layers/kernels are differentiable w.r.t. the shift parameters via bilinear interpolation. The paper then explores what the effect is of changing the standard dense convolution kernels, with the sparse kernels that are able to grow to bigger receptive field sizes while training. Results show that learning the kernel sizes during training is beneficial, whilst typically being more parameter efficient (due to the sparse parametrization), at the cost of only a moderate computational overhead.

**Summary Of The Review:**

Not particularly novel, but the value of the paper is in its experimental work and code release, which seems to be properly documented. I cannot recommend accept as the first part of the paper is sloppy and it isn't very clear what the paper adds relative to existing works on kernels in adaptive bases.

---

> ### Author Response · Authors · 2022-11-16
> **Answer to reviewer 2NcJ - Part 1**
>
> The authors thank the reviewer 2NcJ for his review. In the following, a response to the questions and remarks made by the reviewer:
>
> > The paper then explores what the effect is of changing the standard dense convolution kernels, with the **sparse kernels that are able to grow to bigger receptive field sizes while training**. Results show that **learning the kernel sizes during training is beneficial**, whilst typically being more parameter efficient (due to the sparse parametrization), at the cost of only a moderate computational overhead.
>
> This is not what our paper claims, in DCLS we are not learning kernel sizes but rather the position of every kernel element inside the kernel limit. Furthermore, the maximum receptive field size is fixed (17 x 17 in our case) and the positions of kernel elements are learned within this limit. We invite the reviewer to reread the two last paragraphs of the Section 1 "Introduction" (the one beginning with "Instead of having a grid of kernel element ...") and to look at Figure 2 to see that the dilated kernel size is fixed while the kernel count might change.
>
> ### 1. The presentation of the bilinear interpolation is really confusing and possibly incorrect.
>
> Several reviewers pointed out that Section 2 (Kernel Construction in DCLS) was not very clear. This is due to the fact that the notations and preliminaries had been moved  to appendix 7 to save space. We understood that this might have affected the readability of the paper, so we moved the notations and preliminaries back to Section 2 in this new version, while refactoring this section to make it more understandable.
>
> > Equation 1 defines the kernel in terms of a function f **but the function f is nowhere explicitly defined** . The definition is somehow implicit in equation 3.
>
> The function $f$ was defined in Eqs. (2) and (3) in the previous version of the manuscript, now Eqs. (3) and (4) in the new version. It is the function that defines the DCLS kernel in the scalar case of DCLS. Function $F$ is the same but in the general case.
>
> > $p^1$ and $p^2$ are not defined.
>
> A definition for $p^1$ and $p^2$ was added in Subsection 2.1 Notations and preliminaries. As inputs to the function $F$, they are defined as vectors. We respected the guidelines of the conference which stipulate that vectors should be written in bold characters, while non-bold characters $p^1$ and $p^2$ are the positions in the scalar case.
>
> > $n$, is not defined, nor is $p$,
>
> We changed those to $s _1$ and $s _2$, respectively.
>
> > In equation 1 however, index i is used to iterate over the weights/centers of the kernel, wherease in (3) it is used to iterate over the rows of K. This is somewhat confusing.
>
> We changed this silent index to $k$ for better readability.
>
> We invite the reviewer to reread this section in the new version of the paper to see if it is more understandable.
>
> ### 2. Recent works on learnable receptive field sizes seem to be missing
> In Section 5, we now cite all the related work suggested by reviewer 2NcJ and compare their approaches to ours. In particular, we compare DCLS to the following methods while highlighting the notable differences between each method and ours:
>
> - Romero, D. W., Kuzina, A., Bekkers, E. J., Tomczak, J. M., \& Hoogendoorn, M. (2021). Ckconv: Continuous kernel convolution for sequential data.
> - Romero, D. W., Bruintjes, R. J., Tomczak, J. M., Bekkers, E. J., Hoogendoorn, M., \& van Gemert, J. C. (2021). Flexconv: Continuous kernel convolutions with differentiable kernel sizes.
> - Jacobsen, J. H., Van Gemert, J., Lou, Z., \& Smeulders, A. W. (2016). Structured receptive fields in cnns
> - Pintea, S. L., Tömen, N., Goes, S. F., Loog, M., \& van Gemert, J. C. (2021). Resolution learning in deep convolutional networks using scale-space theory.
> - Shelhamer, E., Wang, D., \& Darrell, T. (2019). Blurring the line between structure and learning to optimize and adapt receptive fields.
>
> We also cite:
> - Bekkers, E. J. (2019). B-spline CNNs on lie groups.

---

> > ### Author Response · Authors · 2022-11-16
> > **Answer to reviewer 2NcJ - Part 2**
> >
> > ### Minor details.
> > > It would be nice to give a bit more detail on what depthwise implicit gemm convolution entails. It is mentioned several times and a high level description could help the reader without them having to look up the original paper.
> >
> > More details about this method have been added to the last item (Depthwise implicit gemm) of the list in Section 3 Learning techniques.
> >
> > > Regarding the position initialization (section 3). "In an attempt to facilitate learning, we chose an initial distribution close to the one obtained at the end of training", is this reasonable?
> >
> > In our opinion, it was worth trying, but as said in the manuscript, it did not help: a uniform distribution gives a similar final accuracy.
> >
> > > What if the initial distribution was chosen poorly, then also the one at the end of training would be right? ?
> >
> > If the initial distribution is chosen as uniform for example, we notice that the final distribution of positions is also close to a centered normal law of std 0.5. So the initial distribution does not seem to matter much.
> >
> > > "Yet in practice, ..." so would you recommend using uniform initialization then instead
> >
> > We do not have a recommendation. By default, we use the normal law with a 0.5 std.
> >
> > > On "Dilated kernel size tuning" I can imagine that the overhead **could be independent from the constrained kernel size**. Since convolution and kernel construction are linear, one could do the interpolation on the input feature maps instead (for the m different shifts) and then simply combine the results with a linear layer afterwards.
> >
> > The overhead is dependent of the dilated kernel size (bigger sizes lead to greater overheads), a timing measurement study has been added in appendix 13 to further emphasize this.
> >
> > The proposed implementation using an interpolation of the input along with a linear layer afterwards could work, but it implies the construction of the interpolation for every weight of the kernel which could be limiting in terms of computing time.
> >
> >
> > > In related work the effective RF size is discussed and the reader is referred to appendix 10. The idea of looking at the ERFs is interesting, however, app 10 does not give much detail on how the "heatmaps" are obtained.
> >
> > ERFs are obtained by finding (via backpropagation) the input pixels that most impact the activations of a pretrained model using samples of the dataset. The ERFs presented in the appendix were obtained by using and adapting this code : <https://github.com/DingXiaoH/RepLKNet-pytorch/blob/main/erf/visualize_erf.py>.
> >
> >
> > > When discussin the method in comparison to deformable convolutions, it mentions "Firstly, in deformable convolutions, the offsets ..." I do not consider this to be a difference. Parametrizing the locations as offsets rather then absolute points (which are offsets to an origin) boils down to the same thing.
> >
> > The reviewer is right about this, so we removed this sentence from the paper.
> >
> > > Also, the "secondly", it indeed describes a difference, but I consider **dilated convs** to be strictly more general than the proposed work (i.e. one could decide with deformable convs not to make the kernels dependent on the input)
> >
> > Maybe the reviewer wanted to say that the deformable convolution (and not the dilated convolution) is more general than DCLS. If it is the case, then the answer is yes, as framed by reviewer N17u "The paper shows that the convolution coordinates in dilated convolution can be more flexible than a fixed, grid structure. In this sense, this work showcases the middle point between dilated convolution and deformable convolution (the latter has the added flexibility being input-dependent)."

---

> > > ### Comment · Reviewer_2NcJ · 2022-11-21
> > > **thank you for the clarifications**
> > >
> > >    > This is not what our paper claims, in DCLS we are not learning kernel sizes but rather the position of every kernel element inside the kernel limit. Furthermore, the maximum receptive field size is fixed (17 x 17 in our case) and the positions of kernel elements are learned within this limit. We invite the reviewer to reread the two last paragraphs of the Section 1 "Introduction" (the one beginning with "Instead of having a grid of kernel element ...") and to look at Figure 2 to see that the dilated kernel size is fixed while the kernel count might change.
> > >
> > > Thank you for the invite, I did, and I don't it was necessary as I believe to have understood correct upon first reading. My summary was not really accurate though, and my appologies for that. I do think however that even though you set the max kernel size (to 17x17), this does not necessarily say anything about the effective receptive field size. It is the spread of the centers that does, your method could learn to spread out the centers more during training. My statement was to limiting and should have said something about "learning the locations of the weights" during training helps.
> > >
> > >    > The function  was defined in Eqs. (2) and (3) in the previous version of the manuscript, now Eqs. (3) and (4) in the new version. It is the function that defines the DCLS kernel in the scalar case of DCLS. Function  is the same but in the general case.
> > >
> > > f is still not defined. Equation 2 only give the overall form of the function F (capital F?) not a definition, it does give a definition of K in terms of f, but f still not defined. Again, equation 2,3 and 4 together might be used to infer what f does with some inverse engineering. But the whole presentation is just not precise and to the point and thus I still believe the presentation of interpolation is rather poor.
> > >
> > > Regarding related work; thanks for discussing it. However, there's one statement in the paper that bothers me:
> > >
> > >    > In addition, they cannot be used as a drop-in replacement for the depthwise separable convolution in ConvNeXt, at least in their
> > > current forms, so we could not benchmark them with DCLS.
> > >
> > > I do not understand this argument of considering methods as "drop-in" replacement. I.m.o., all those methods are as "drop-in" as the presented layer in this paper. I do not understand why they are not considered as such. In fact, I believe several of those works design methods in terms of ConvBlocks, be it maybe more of the resnet type, which was the big thing before ConvNext.
> > >
> > > As for the rest, thank you for your clarifications.

---

> > > > ### Author Response · Authors · 2022-11-21
> > > > **Thank you for your response**
> > > >
> > > > The authors would like to thank reviewer 2NcJ for his response.
> > > >
> > > > > it does give a definition of K in terms of f, but f still not defined.
> > > >
> > > > The definition of f is mathematically correct. f is defined as a function that takes as input 3 real variables and returns a matrix K of size s1 x s2 that is made of zeros except for at most 4 adjacent components that represent the 2D interpolation of the single weight w as described by (3), (4) and (5). At this point, the authors do not know how to make this definition more explicit.
> > > >
> > > > > Again, equation 2,3 and 4 together might be used to infer what f does with some inverse engineering. But the whole presentation is just not precise
> > > >
> > > > Could reviewer 2NcJ provide a better way to write this definition ? Maybe the other reviewers and the AC could give their opinion ?
> > > >
> > > > > I do not understand this argument of considering methods as "drop-in" replacement. I.m.o., all those methods are as "drop-in" as the presented layer in this paper. I do not understand why they are not considered as such.
> > > >
> > > > The main reason is that those methods are not compatible with the depthwise separable convolution.
> > > >
> > > > > I believe several of those works design methods in terms of ConvBlocks, be it maybe more of the **resnet** type, which was the big thing before **ConvNext**.
> > > >
> > > > A major difference between ResNet and ConvNeXt is the depthwise separable convolution which significantly reduces the number of parameters and the throughput of the model. Non-separable convolution and thereafter the methods mentioned in the related work could be adapted in their current state to ConvNeXt but this will have to be accompanied by a modification of the model architecture on the macro scale and the convolution blocks will have to change (channels, depth ...) in order to stick to the parameter count and the throughput. Ultimately, by doing so, we will no longer have a ConvNeXt architecture but a brand new one. This is why for example deformable convolution v2 is not suited "as a drop-in replacement of the depthwise separable convolution in ConvNext", and thus motivated the development of deformable convolution v3 which further improves ConvNeXt and for which work is still in progress at the time of writing this response. As said in the paper, we have already tried to implement a depthwise separable version for deformable convolution v2 without success.

---

### Official Review · Reviewer_zKnb · 2022-10-30

**Confidence:** 5
**Correctness:** 3
**Technical Novelty And Significance:** 2
**Empirical Novelty And Significance:** 2
**Recommendation:** 8

**Clarity, Quality, Novelty And Reproducibility:**

The paper is clear. The quality of writing is high. And the authors did everything to facilitate reproducibility

**Strength And Weaknesses:**

The motivation of the paper is clear. It is well-understood both from the previous experience, from the related work and from the results presented in the paper, that larger receptive fields allow for more accurate CNNs in some scenatrious.

**Strengths**
1. The paper is well-written. The story is coherent and it is easy to follow. The main results are presented in an understandable form.
2. The authors share the implementation of the algorithm which is a huge plus as it allows one to understand it better.

**Weaknesses**
The main weakness of the current manuscript is that it does not consider a huge part of the related work. I am sure that the following papers should be taken into consideration. At least, they should be mentioned. It will be a huge plus to my current rating if the proposed method is somehow compared to these papers
1. In [1] the authors propose to decompose filters as gaussian derivatives and vary their scales. Thus the model is able to utilize larger filters with the same number of trainable parameters. Their results demonstrate that such a method significantly improves the models' performance.
2. In [2, 3] the authors propose to reparametrizer convolutional filters and learn their parameters. Which is very close to the proposed method in its spirit
3. In [4, 5, 6, 7, 8, 9] the authors buid scale-scale equivariant neural networks which focus aroung the scale problem and propose everal interesting methods for a problem similar to the considered
4. In [4, 6, 7] the authors use dilation and an alternative of dilation for fractional spacing, which is close to the proposed idea, although the initial motivation may vary.

- [1] Jacobsen J. H. et al. Structured receptive fields in cnns.
- [2] Romero D. W. et al. Flexconv: Continuous kernel convolutions with differentiable kernel sizes
- [3] Romero D. W. et al. Ckconv: Continuous kernel convolution for sequential data
- [4] Worrall D., Welling M. Deep scale-spaces: Equivariance over scale
- [5] Sosnovik I., Szmaja M., Smeulders A. Scale-equivariant steerable networks
- [6] Sosnovik I., Moskalev A., Smeulders A. Disco: accurate discrete scale convolutions
- [7] Sosnovik I., Moskalev A., Smeulders A. How to Transform Kernels for Scale-Convolutions
- [8] Bekkers E. J. B-spline cnns on lie groups
- [9] Zhu W. et al. Scaling-Translation-Equivariant Networks with Decomposed Convolutional Filters


**Summary Of The Paper:**

The paper considers the problem of how to allow for larger receptive fields for better accuracy with affordable compuitational complexity. The authors propose an algorithm based on learnable spacing for dilated convolutions.

**Summary Of The Review:**

The current version of the manuscript misses several very important papers which have a similar spirit while the motivation may slightly vary. Understanding the significance of paper without comparing it to other approaches is not possible.

===================

While the intiial rating was 6, I adjusted it to 8. The authors improved the paper according to my recommendations.

---

> ### Author Response · Authors · 2022-11-15
> **Answer to reviewer zKnb**
>
> The authors thank the reviewer zKnb for his review. In the following, a response to the remark made by the reviewer:
> > The main weakness of the current manuscript is that it does not consider a huge part of the related work. I am sure that the following papers should be taken into consideration. At least, they should be mentioned. It will be a huge plus to my current rating if the proposed method is somehow compared to these papers
>
> In section 5, we now cite all the related work suggested by reviewer zKnb and compare their approaches to ours. In particular, we compare DCLS to the following methods while highlighting the notable differences between each method and ours :
>
> - Romero, D. W., Kuzina, A., Bekkers, E. J., Tomczak, J. M., \& Hoogendoorn, M. (2021). Ckconv: Continuous kernel convolution for sequential data.
> - Romero, D. W., Bruintjes, R. J., Tomczak, J. M., Bekkers, E. J., Hoogendoorn, M., \& van Gemert, J. C. (2021). Flexconv: Continuous kernel convolutions with differentiable kernel sizes.
> - Jacobsen, J. H., Van Gemert, J., Lou, Z., \& Smeulders, A. W. (2016). Structured receptive fields in cnns.
>
>
>
> We also cite the different papers related to scale-equivariant methods:
>
> - Worrall, D., \& Welling, M. (2019). Deep scale-spaces: Equivariance over scale.
> - Sosnovik, I., Szmaja, M., \& Smeulders, A. (2019). Scale-equivariant steerable networks.
> - Sosnovik, I., Moskalev, A., \& Smeulders, A. (2021). Disco: accurate discrete scale convolutions.
> - Sosnovik, I., Moskalev, A., \& Smeulders, A. (2021). How to Transform Kernels for Scale-Convolutions.
> - Bekkers, E. J. (2019). B-spline cnns on lie groups.
> - Zhu, W., Qiu, Q., Calderbank, R., Sapiro, G., \& Cheng, X. (2019). Scaling-Translation-Equivariant Networks with Decomposed Convolutional Filters.

---

> > ### Comment · Reviewer_zKnb · 2022-12-03
> > **Improved rating**
> >
> > The paper looks more solid after the modifications. I have adjusted my rating to 8.

---

### Author Response · Authors · 2022-11-12
**Summary of the revision**

We sincerely thank all the reviewers for their valuable comments, which helped us improve the quality of our paper. In light of these comments, we have modified the article, and we summarize the revision in the updated version as follows:
- In the abstract, we added that the main focus of the paper is on the
2D case for computer vision only, to emphasize the fact that the paper will deal only with vision tasks.
- Section 2 was not made very clear to the reviewers. This is due to the fact that the notations and preliminaries have been moved to appendix 7 to save space. We understand that this may affect the readability of the paper, so we have put the notations and preliminaries back in section 2 for this version, while refactoring this section to make it more understandable.
- The pseudo-code algorithms have been moved to appendix 8 to free space for new section 2.
- In section 3, we add more discussion  and details about the dilated kernel size tuning, position synchronizing and the depthwise implicit gemm method.
- In section 4, tables 3 and 4, we add the inference throughput for the downstream tasks (semantic segmentation and object detection) for ConvNeXt and ConvNeXt-dcls models.
- In section 5, we cite all the related work suggested by the reviewers and compare the approaches to ours. We also explain that we have tested the deformable convolution method as a drop-in replacement of the depthwise separable convolution in the ConvNeXt architecture, but this last didn't work as the training becomes very slow (the throughput is divided by 4 for the ConvNeXt-tiny model), in addition to a growing training loss at the first epochs, which means that the method is not adapted as a drop-in replacement of the depthwise separable convolution in the ConvNeXt architecture. Furthermore, we insist again on the differences between this (deformable convolution) and our approach, as only reviewer "N17u" seems to have grasped the major difference between the two.
- We add the actual PyTorch code of the 1D, 2D and 3D kernel construction of DCLS in Appendix 9. This code is written with PyTorch modules only and does not require any additional compilation or library. The code equivalent to this one, written in C++/CUDA language can be found in the Github repository associated with this paper.
- In appendix 13, we add bar charts where we compare the execution time of a forward + backward pass on the variants of the ConvNeXt model (tiny, small and base) with and without the DCLS method.
- References to the possible use of DCLS in 1D tasks such as audio and 3D tasks such as video classification have been added in the conclusion.

**A detailed answer to each of the reviewers' questions will follow in the next few days.**

---

### Decision · Program_Chairs · 2023-01-20

**Decision:**

Accept: poster

**Justification For Why Not Higher Score:**

Open question about how much this method will be used in the real world. Niche sub-field.

**Justification For Why Not Lower Score:**

Solid experiments. Parsimonious idea.

**Metareview: Summary, Strengths And Weaknesses:**

In this work, the authors propose a method to improve learning with convolutional filters on computer vision problems. In particular, the authors describe a method that learns location of each convolutional filter by treating the spacing between filter elements as a learnable hyperparameter. The position of each kernel pixel is learnable and differentiable through the usage of linear interpolation. The authors demonstrate the efficacy of their method on image classification with ImageNet-1K and show favorable gains in performance given a particular computational or parameter budget. Additionally, the authors empirically verified the efficacy of these kernels on semantic segmentation with ADE-20K, object detection with COCO and transfer learning/robustness evaluation on variations of ImageNet, again demonstrating reasonable gains in performance given the moderate cost of these kernels.

The reviewers positively commented on the clarity of presentation, the sharing of the code, the simplicity of the method, and the extensive experiments (including some negative results which this AC especially applauds). The reviewers also identified several weaknesses.
* One reviewer commented on the lack of experiments on audio where dilated kernels are extremely important. Upon my review and discussion with the reviewers, although it would be nice to see such results, the promising experiment with vision provides enough reason to consider applying this to other domains. Accordingly, I concur with the authors that the lack of experiments in audio is not a reason for rejection.
* One reviewer highlighted some strong concerns about the strength of the results (e.g. marginal gains on ImageNet) and questioned whether this method would be used given that customized kernels must be employed on a given hardware architecture. Furthermore, other methods (e.g. ConvNext) offer other, simpler methods for achieving larger convolutional kernels. The authors responded that the gains are appreciable especially compared to the parameter and computational burden of the proposed method. As a point of comparison, the authors showcased the differential between the “small” and “large” versions of ConvNext not offering comparable gains in performance given the increased burden of the larger model.
* Multiple reviewers also commented on several missing citations and comparisons with other pieces of literature related to learnable versions of convolutional filters. One point of clarification that arose in discussion was the comparison of this work to [Deformable Convolutions](https://arxiv.org/abs/1703.06211). This discussion elucidated that the proposed method produces kernels that are not dependent on the input, unlike the previous method. Although the authors added missing citations to the paper accordingly, there is still an open question about how much of a contribution this work provides above and beyond previous methods. At the minimum, the authors are encouraged to strongly revise their motivation section justifying the need for this method in light of other methods?

Given all of the positive and negative aspects of the paper, this paper was considered borderline. Accordingly, I led an online discussion with 4 of the reviewers on the merits of the paper being accepted. There was general consensus that this paper was well written but there was an open question about the overall motivation of this work above and beyond well-researched literature on “learnable” versions of convolutional kernels. That said, reviewers all concurred that this is a solid paper in terms of presentation and the soundness of the technique approaching a problem with a long history in computer vision. For all of these reasons, we have decided to accept this paper into this conference so long as the author updates their manuscript to address all of the issues surfaced in the review period and provide a notable revision of the motivation section.


**Note From Pc:**

if the above contains the word "oral" or "spotlight" please see: "oral" presentation means -> notable-top-5% and "spotlight" means -> notable-top-25%. As stated in our emails, we are disassociating presentation type from AC recommendations

**Summary Of Ac-Reviewer Meeting:**

Discussion. 4 Participants.
- Consensus that the paper was well written.
- Open question about how this work relates to the large literature of deformable/dilated convolutions. That said, the contrast with Deformable Convolution was clear.
- Open question about how much this method will be used in the real world, but there was also a sentiment that this a good paper that will contribute to the field of ideas.
- One of the largest issues of concern was the motivation. What exactly is this method solving above and beyond other methods?
- Consensus that this is not an oral presentation but this is solid work.

---

> ### Author Response · Authors · 2023-02-13
> **Summary of the camera ready revision**
>
> The authors would like to sincerely thank the AC and all reviewers for their helpful suggestions and comments throughout the review process. Based on the meta-review comments, we have made the following changes to the camera ready version:
> - As requested by the reviewers, the motivation of the work has been emphasized in the introduction. To that end, the authors have reworked the fifth paragraph of the introduction (the one that now starts with “The principal motivation of DCLS is …”).
> - Camera-ready version have been turned on, as a consequence authors names and affiliations have been added at the very beginning of the paper.
> - All anonymous links were changed to non-anonymous ones and moved to the reproducibility statement to save space. Since the authors' names shifted the conclusion to the 10th page, this last modification permitted to return to the required length of 9 pages.
> - In the first paragraph of the related work, the authors added this sentence :
> " Secondly, DCLS positions are channel-dependent, unlike deformable convolution ones.” as an extra difference between the DCLS method and the deformable convolution one in order to convey that DCLS certainly does not allow input-dependent learning, but it allows channel-dependent position learning instead. And conversely, deformable convolution allows input-dependent learning at the expense of channel-dependent position learning.